# Regulation of bone homeostasis by MERTK and TYRO3

Janik Engelmann[1,2,3,4,5], Jennifer Zarrer[6,7,8], Victoria Gensch[1,2,3,4,5], Kristoffer Riecken [9], Nikolaus Berenbrok[1,2,3,4,5], The Vinh Luu[1,2], Antonia Beitzen-Heineke[1,2], Maria Elena Vargas-Delgado[1,2,3,4,5], Klaus Pantel [2], Carsten Bokemeyer[1], Somasekhar Bhamidipati[10], Ihab S. Darwish[10], Esteban Masuda[10], Tal Burstyn-Cohen [11], Emily J. Alberto[12], Sourav Ghosh [13,14], Carla Rothlin [12,13], Eric Hesse [7,8], Hanna Taipaleenmäki[6,7,8], Isabel Ben-Batalla[1,2,3,4,5,15] ✉ & Sonja Loges [1,2,3,4,5,15] ✉

The fine equilibrium of bone homeostasis is maintained by bone-forming osteoblasts and bone-resorbing osteoclasts. Here, we show that TAM receptors MERTK and TYRO3 exert reciprocal effects in osteoblast biology: Osteoblast-targeted deletion of MERTK promotes increased bone mass in healthy mice and mice with cancer-induced bone loss, whereas knockout of TYRO3 in osteoblasts shows the opposite phenotype. Functionally, the interaction of MERTK with its ligand PROS1 negatively regulates osteoblast differentiation via inducing the VAV2-RHOA-ROCK axis leading to increased cell contractility and motility while TYRO3 antagonizes this effect. Consequently, pharmacologic MERTK blockade by the small molecule inhibitor R992 increases osteoblast numbers and bone formation in mice. Furthermore, R992 counteracts cancer-induced bone loss, reduces bone metastasis and prolongs survival in preclinical models of multiple myeloma, breast- and lung cancer. In summary, MERTK and TYRO3 represent potent regulators of bone homeostasis with cell-type specific functions and MERTK blockade represents an osteoanabolic therapy with implications in cancer and beyond.

Bone represents a dynamic tissue constantly renewed and reshaped by osteoclastic bone resorption and osteoblastic bone formation. The process of bone formation is dependent on osteoblasts[1]. Their function is tightly regulated by a signaling pathway network controlling differentiation from mesenchymal stem cell-derived osteoprogenitor cells via osteoblasts towards osteocytes[2]. For example, the WNT signaling pathway induces osteoblast differentiation via the binding of WNT ligands to LRP5/6[3]. The discovery of the WNT antagonist Sclerostin led to the development of anti-sclerostin antibodies (Romosozumab), which were recently approved in the indication of postmenopausal osteoporosis in the United States and Europe. Despite its success, the bone anabolic effect of

Romosozumab is short-lived, and therapy needs to be followed by established antiresorptive agents to reach long-term effects in osteoporosis[4–6]. Furthermore, it is associated with potential adverse cardiac events[5]. The investigation of additional pathways controlling osteoblast differentiation and function is therefore essential to improve osteoanabolic treatments.

The TAM family of receptor tyrosine kinases, consisting of TYRO3 (BRT, DTK, RSE, SKY, and TIF), AXL (ARK, TYRO7, and UFO), and MERTK (EYK, NYM, and TYRO12) and their cognate ligands growtharrest-specific gene-6 (GAS6) and protein S (PROS1) represent cell surface transmembrane receptors. They trigger phosphorylation and activation of multiple downstream signaling proteins influencing

tissue homeostasis in several organ systems by modulating key processes, including tissue repair, inflammation, cell survival, proliferation, and migration[7–13]. The GAS6-TYRO3 axis induces osteoclast differentiation and it was consistently shown that germline deletion of *TYRO3* leads to increased bone mass[14]. However, the tissue-specific roles of TAM receptors in the bone, especially in osteoblasts, are unknown.

TAM receptors and their ligands PROS1 and GAS6 are frequently overexpressed in cancer and mediate tumor-stroma interaction to limit anti-tumor immunity and fuel cancer growth[15,16]. Bone is a major site of cancer metastases, often presenting as osteolytic bone metastases[17]. Osteolytic bone metastases lead to skeletal-related events (SREs), including pathological fractures and pain, that require palliative interventions[18]. When established in the bone, cancer cells secrete factors, which educate osteoblasts to promote tumor growth and suppress their bone-forming capacity[19]. Several studies hypothesize that restoring osteoblast function could mitigate metastatic osteolytic bone disease[20–22].

In this work, we demonstrate that osteoblasts are controlled by the TAM receptors MERTK and TYRO3, which regulate key cellular functions, including differentiation, migration, and bone formation. Furthermore, we show that cancer cells exploit this regulatory mechanism and aggravate osteolytic bone disease via MERTK. Treatment of bone metastasis-bearing mice with a MERTK-targeting small molecule inhibitor alleviated bone destruction and prolonged life span in mice, suggesting MERTK blockade as an osteoanabolic therapy in osteopenic bone diseases.

## Results

### Expression of TAM receptors in osteoblasts

To gain insight into the role of TAM receptors in osteoblast biology, we performed gene expression analysis of *Mertk*, *Tyro3*, *Axl*, *Gas6*, and *Pros1* mRNA in primary murine calvarial cell osteoblast cultures. Upon osteogenic induction, we observed an upregulation of *Mertk*, *Gas6*, and *Pros1* during the culture period of 3 weeks, whereas *Axl* was downregulated. *Tyro3* showed a short peak of twofold enhanced expression in the early differentiation phase with constant expression levels thereafter (Fig. 1a).

### Conditional deletion of *Mertk* and *Tyro3* in osteoblasts

The increased expression levels of *Mertk* and *Tyro3* led us to investigate their function in osteoblasts. Therefore, we created an osteoblast-targeted knockout of *Mertk* and *Tyro3* in mice. We used the 2.3-kb mouse collagen type I, alpha 1 (Col1a1) promotor for cre recombinase expression (Col1a1-cre⁺), which is well described for being exclusively expressed in osteoblast lineage and odontoblasts without significant leakage to other tissues[23]. We deleted *Mertk* and *Tyro3* by crossing *Col1a1-cre⁺* mice with *Mertk^{flox/flox}* mice[1] and *Tyro3^{flox/flox}* mice (SFig. 1a, b), generating *Col1a1-cre⁺;Mertk^{flox/flox}* and *Col1a1-cre⁺;Tyro3^{flox/flox}* conditional knockout mice (referred to as *Mertk−/−^{OB}* and *Tyro3−/−^{OB}*). *Col1a1-cre⁻;Mertk^{flox/flox}* and *Col1a1-cre⁻;Tyro3^{flox/flox}* littermates were used as controls throughout the experiments. *Mertk−/−^{OB}* and *Tyro3−/−^{OB}* mice exhibited no obvious skeletal defects and normal tooth eruption. However, whereas body weight in *Mertk−/−^{OB}* (SFig. 2a) was not altered, *Tyro3−/−^{OB}* mice showed slightly decreased body weight in comparison

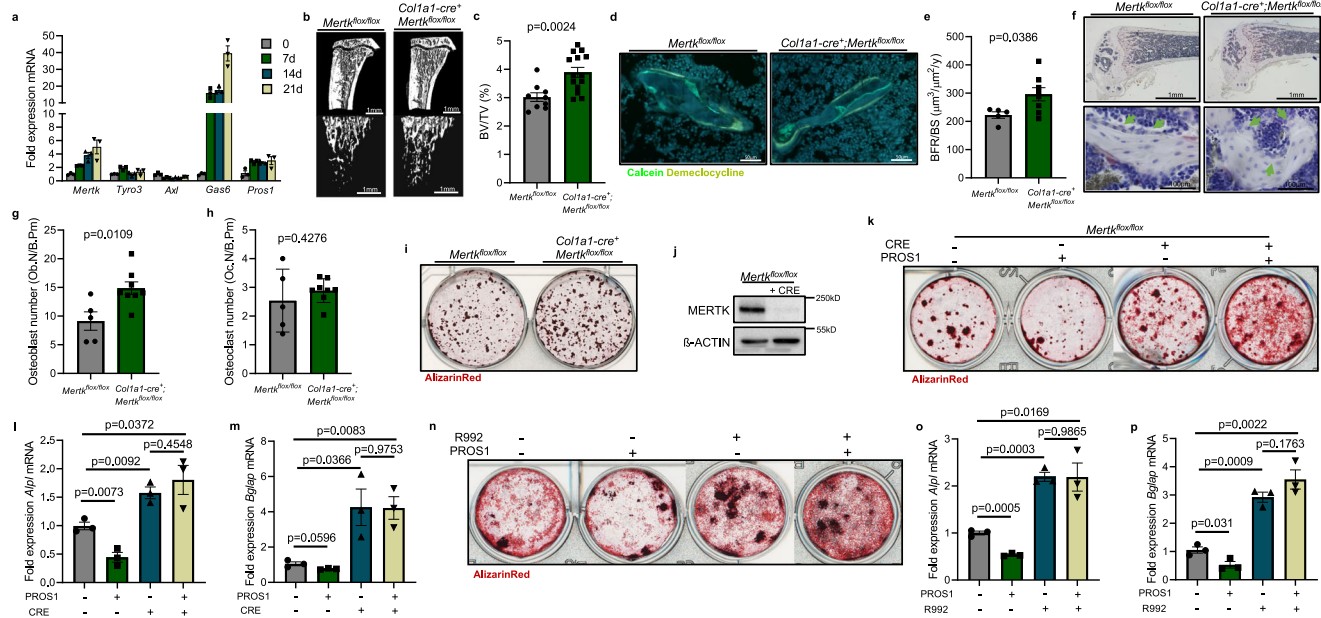

**Fig. 1 | Negative regulation of bone formation by TAM receptor MERTK. a** mRNA expression analysis of TAM receptor family *Mertk*, *Tyro3*, *Axl*, *Gas6*, and *Pros1* in primary murine osteoblast cultures after 0, 7, 14, and 21 days (*n* = 3 biological replicates) (*Mertk* expression d0 vs. d7: *p* = 0.0002; d0 vs. d14: *p* = 0.0042; d0 vs. d21: *p* = 0.0172), (*Tyro3* expression d0 vs. d7: *p* = 0.009; d0 vs. d14: *p* = 0.9604; d0 vs. d21: *p* = 0.2928), (*Axl* expression d0 vs. d7: *p* = 0.0161; d0 vs. d14: *p* = 0.0152; d0 vs. d21: *p* = 0.0331), (*Gas6* expression d0 vs. d7: *p* = 0.0011; d0 vs. d14: *p* = 0.0002; d0 vs. d21: *p* = 0.001), (*Pros1* expression d0 vs. d7: *p* = 0.0157; d0 vs. d14: *p* = 0.0181 d0 vs. d21: *p* = 0.054). **b, c** Microcomputed tomography (µCT) of the metaphyseal proximal region of tibias from 8-week-old *Mertk^{flox/flox}* and *Col1a1-cre⁺;Mertk^{flox/flox}* female mice (top, longitudinal view of cortical and cancellous bone; bottom, longitudinal view of cancellous bone) (**b**). Quantification of bone volume (BV/TV) of cancellous bone determined by µCT analysis (**c**) (*Mertk^{flox/flox}*, *n* = 9; *Col1a1-cre⁺*; *Mertk^{flox/flox}*, *n* = 14). **d, e** Representative pictures of Calcein Demeclocycline labeling of *Mertk^{flox/flox}* and *Col1a1-cre⁺;Mertk^{flox/flox}* female mice (**d**). Bone formation rate of

*Col1a1-cre⁺;Mertk^{flox/flox}* mice after 8 weeks (**e**) (*Mertk^{flox/flox}*, *n* = 5; *Col1a1-cre⁺*; *Mertk^{flox/flox}*, *n* = 8). **f–h** Representative pictures (**f**) and histomorphometric analysis of osteoblast (**g**), and osteoclast number (**h**) by TRAP/Hematoxylin staining in femur from *Col1a1-cre⁺;Mertk^{flox/flox}* mice. Green arrows pointing to osteoblasts visible as cuboidal or polygonal mononuclear cells on the endosteal bone surface (*Mertk^{flox/flox}*, *n* = 5; *Col1a1-cre⁺;Mertk^{flox/flox}*, *n* = 8). **i** Alizarin Red staining of ex vivo calvarial cell osteoblast culture from *Col1a1-cre⁺;Mertk^{flox/flox}* mice. **j** Analysis of MERTK protein in osteoblast cultures from *Mertk^{flox/flox}* mice treated with recombinant CRE recombinase. ß-ACTIN run on a separate gel. **k** Alizarin Red staining of MERTK KO calvarial cell cultures treated with PROS1 (100 nM). **l, m** RT-qPCR analysis of *Alpl* (**l**) *and Bglap* (**m**) mRNA expression in MERTK KO osteoblasts (*n* = 3 biological replicates). **n** Alizarin Red staining of wild-type calvarial cell cultures treated with PROS1 (100 nM) and MERTK-inhibitor R992 (200 nM). **o, p** RT-qPCR analysis of *Alpl* (**o**) *and Bglap* (**p**) mRNA expression (*n* = 3 biological replicates). Data were means ± SEMs. Statistical significance was determined by a two-tailed unpaired *t*-test.

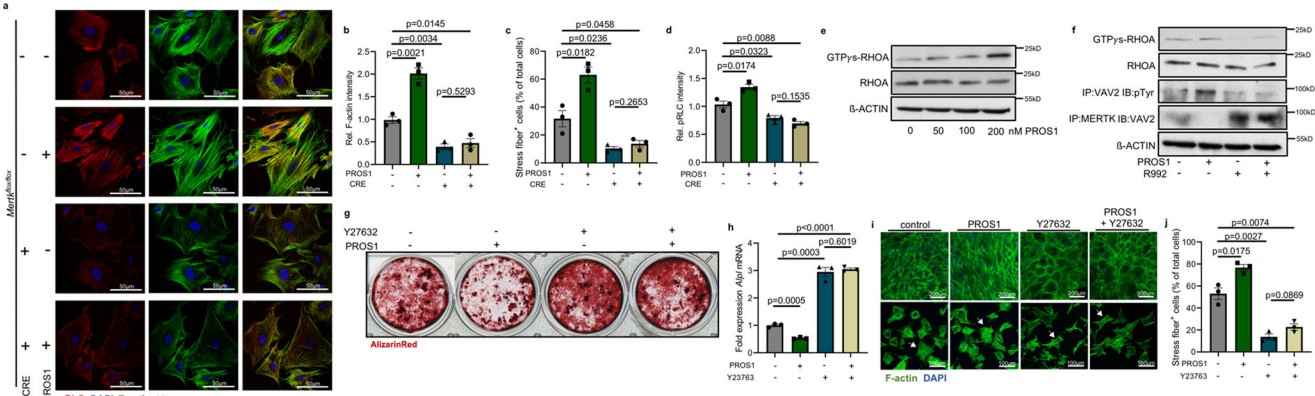

**Fig. 2 | Inhibition of bone formation by MERTK is mediated by PROS1-MERTK-VAV2-RHOA-ROCK axis. a–d** Confocal imaging of pRLC and F-actin staining of MERTK KO osteoblasts on glass coverslips treated with TAM receptor ligand PROS1 (100 nM) (**a**). F-actin intensity (*n* = 3, mean of 50 measurements in three fields) (**b**), stress fiber-containing cells (*n* = 3, mean of 100 measurements in three fields) (**c**), and pRLC intensity (*n* = 3, mean of 50 measurements in three fields) (**d**) was quantified. **e** Immunoblot of wild-type osteoblasts treated with different concentrations of PROS1 (0, 50, 100, 200 nM) showing activated GTPγs-bound RHOA, total-RHOA, and ß-ACTIN. ß-ACTIN run on a separate gel. **f** Immunoblot of wild-type osteoblasts treated with PROS1 (100 nM) and MERTK-inhibitor R992 (200 nM) showing activated GTPγs-bound RHOA, total-RHOA, phosphorylated VAV2, an

association of MERTK and VAV2 and ß-ACTIN. ß-ACTIN run on a separate gel. **g** Alizarin Red staining of wild-type calvarial cells treated with PROS1 (100 nM) and ROCK-inhibitor Y27632 (10 µM). **h** RT-qPCR analysis of *Alpl* mRNA expression (*n* = 3 biological replicates). **i** Calvarial cells were cultured in an osteogenic medium with exogenous addition of PROS1 and Y27632. Cultures were stained by F-actin staining. The top view shows representative pictures of the cell morphology of calvarial cell cultures after 5 days. The bottom view shows representative pictures of single-cell analysis on fibronectin substrate (representative image section). **j** The percentage of stress fiber-containing cells was quantified (*n* = 3, mean of 100 cells in three fields). Data were means ± SEMs. Statistical significance was determined by a two-tailed unpaired *t*-test.

to littermate controls (SFig. 2b). Quantitative RT-PCR analysis of mRNA extracted from ex vivo osteoblast cultures confirmed decreased expression of *Mertk* (SFig. 3a) and *Tyro3* (SFig. 3b), showing efficient deletion comparable to in vitro expression of other genes e.g. including Smad1 under the Col1a1 promoter[24].

## Role of MERTK in osteoblasts

Microcomputed tomography (µCT) analysis of the metaphyseal tibia of 8-week-old *Mertk−/−OB* mice in comparison to controls showed an increase in bone volume (+28.9%) (Fig. 1b, c), trabecular number (+21.8%) (SFig. 4a) and trabecular thickness (+5.8%) (SFig. 4b), whereas trabecular separation was decreased (−22.2%) (SFig. 4c). Midshaft evaluation of femurs of *Mertk−/−OB* mice did not show significant changes in cortical bone thickness (SFig. 4d), indicating that *Mertk* might play a more significant role on trabecular bone than on cortical bone. Dynamic histomorphometry showed an increased bone formation rate in *Mertk−/−OB* mice (Fig. 1d, e). Osteoblast numbers in *Mertk−/−OB* mice were increased, whereas osteoclast numbers were not affected (Fig. 1f–h). Ex vivo calvarial cell cultures of *Mertk−/−OB* mice showed increased osteoblast matrix mineralization (Fig. 1i). These data demonstrate that TAM receptor MERTK negatively regulates physiological bone remodeling by inhibition of osteoblastic bone formation.

To further study the role of MERTK in osteoblast differentiation in vitro, we induced stable knockout of *Mertk* using the Cre-Lox recombination technology and delivered recombinant CRE to calvarial cell cultures from *Mertk^flox/flox* mice. Efficient MERTK knockout was evaluated by SDS-PAGE (Fig. 1j). Treatment of calvarial cells with MERTK-ligand PROS1 inhibited osteoblast matrix mineralization in control conditions while inducing it in the absence of MERTK receptor (Fig. 1k). Consistent findings were obtained when measuring alkaline phosphatase (*Alpl*) mRNA expression on day 7 and Osteocalcin (*Bglap*) on day 21: PROS1 decreased expression of these osteoblast differentiation markers, whereas MERTK KO osteoblasts showed increased expression. Notably, PROS1 could not decrease the osteoblast differentiation marker in the absence of MERTK (Fig. 1l, m). Furthermore, the MERTK-specific small molecule inhibitor R992 effectively suppressed the inhibitory effect of PROS1 on matrix mineralization and

differentiation marker expression in wild-type cells (Fig. 1n–p). Altogether, these results show that the PROS1-MERTK axis inhibits osteoblast differentiation and matrix mineralization.

MERTK regulates cell morphology and migration by modulation of actin cytoskeletal rearrangement, primarily via small GTPase RHOA[25], and it was shown that this pathway is detrimental to terminal osteoblast differentiation and bone formation[26].

Therefore, we characterized in a first step osteoblast cell morphology which is mainly regulated by the F-actin cytoskeleton, whose arrangement is constantly renewed to enable different mechanical forces needed for diverse cellular functions such as differentiation, adhesion, or migration[27]. Osteoblast differentiation is associated with low F-actin content and reduced stress fiber formation[28]. Stress fibers are thick crosslinked F-actin bundles promoting cellular retraction via interaction with myosin motor proteins[29,30]. Confocal microscopy of F-actin immunofluorescence staining revealed that PROS1 promoted high F-actin staining intensity and could prominently induce stress fiber formation, whereas MERTK KO osteoblast cultures exhibited low F-actin content and a reduced number of stress fiber-containing cells. PROS1 could not induce stress fiber formation in the absence of MERTK (Fig. 2a–c). As the assembly and function of actomyosin stress fibers depend on myosin regulatory light chain (RLC) phosphorylation[30], we stained for RLC phosphorylation of Ser19, which was reduced in MERTK KO osteoblasts (Fig. 2a, d). These results indicate that MERTK has profound effects on osteoblast cytoskeletal arrangement, with MERTK inducing cellular retraction and stress fiber formation altogether impeding osteoblast differentiation.

Evaluation of RHOA signaling revealed that PROS1 dose-dependently induced activated GTPyS-RHOA in osteoblasts (Fig. 2e). It is known that MERTK can bind SH2 domain proteins, particularly the vav-proto oncogene (vav) family of guanine nucleotide exchange factors (GEFs) for Rho-family GTPases[31,32]. Previous data show that tyrosine phosphorylation of MERTK and VAV leads to VAV dissociation from MERTK and downstream GDP to GTP exchange of RHOA[31]. VAV3 and VAV1 are expressed only in mature osteoblasts, whereas VAV2 is present throughout osteoblast differentiation[33], suggesting that these RHO-GEFS could be involved in the activation of RHOA by MERTK because it binds VAV proteins constitutively. As we observed the

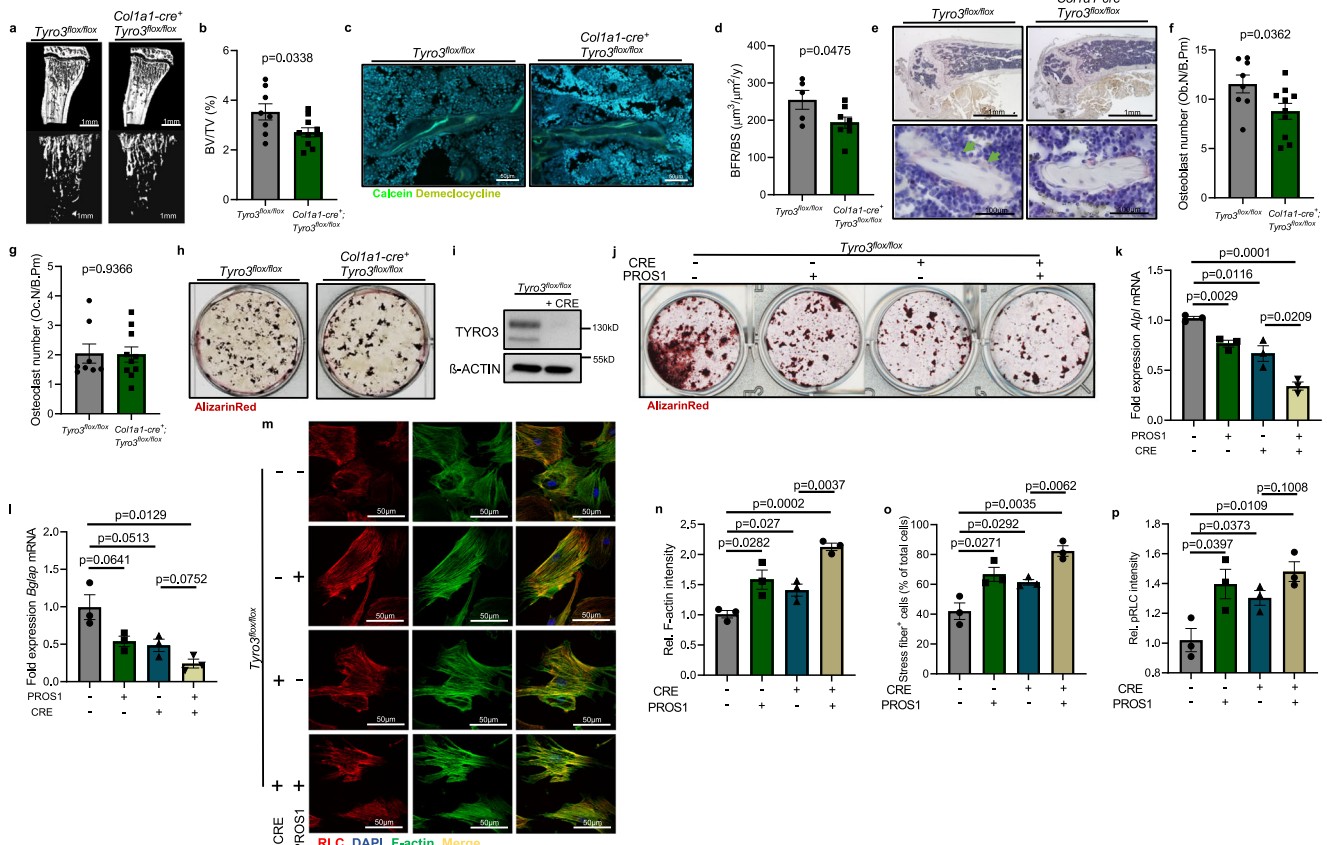

**Fig. 3 | TAM receptor TYRO3 promotes osteoblastogenesis and bone formation. a, b** Microcomputed tomography (μCT) of the metaphyseal proximal region of tibias from 8-week-old *Tyro3flox/flox* and *Col1a1-cre+;Tyro3flox/flox* female mice (top, longitudinal view of cortical and cancellous bone; bottom, longitudinal view of cancellous bone) (**a**). Quantification of bone volume (**b**) of cancellous bone determined by μCT analysis (*Tyro3flox/flox*, *n* = 8; *Col1a1-cre+;Tyro3flox/flox*, *n* = 10). **c, d** Representative pictures of Calcein Demeclocycline labeling of *Tyro3flox/flox* and *Col1a1-cre+;Tyro3flox/flox* female mice (**c**). Bone formation rate of *Col1a1-cre+;Tyro3flox/flox* mice after 8 weeks (*Tyro3flox/flox*, *n* = 5; *Col1a1-cre+;Tyro3flox/flox*, *n* = 8) (**d**). **e**–**g** Representative pictures (**e**) of histomorphometric analysis of osteoblast (**f**) and osteoclast number (**g**) by TRAP/Hematoxylin staining in femur from *Col1a1-cre+;Tyro3flox/flox* mice. Green arrows pointing to osteoblasts visible as cuboidal or polygonal mononuclear cells on the endosteal bone surface (*Tyro3flox/flox*, *n* = 8; *Col1a1-cre+;Tyro3flox/flox*, *n* = 10). **h** Alizarin Red staining of ex vivo calvarial cell osteoblast culture from *Col1a1-cre+;Tyro3flox/flox* mice. **i** Analysis of TYRO3 protein in osteoblast cultures from *Tyro3flox/flox* mice treated with recombinant CRE recombinase. **j** Alizarin Red staining of Tyro3 KO calvarial cell cultures treated with PROS1 (100 nM). **k, l** RT-qPCR analysis of *Alpl* (**k**) *and Bglap* (**l**) mRNA expression in Tyro3 KO osteoblasts (*n* = 3 biological replicates). **m**–**p** Confocal imaging of pRLC and F-actin staining of TYRO3 KO osteoblasts on glass coverslips treated with TAM receptor ligand PROS1 (100 nM) (**m**). F-actin intensity (*n* = 3, mean of 50 measurements in three fields) (**n**), stress fiber-containing cells (*n* = 3, mean of 100 measurements in three fields) (**o**), and pRLC intensity (*n* = 3, mean of 50 measurements in three fields) (**p**) was quantified. Data were means ± SEMs. Statistical significance was determined by a two-tailed unpaired *t*-test.

biological effects of MERTK in all stages of calvarial cell culture, we hypothesized that VAV2 is the main substrate of MERTK, leading to RHOA/ROCK activation. We could demonstrate that blocking of MERTK phosphorylation by R992 promotes high MERTK-VAV2 protein interaction leading to low VAV2 phosphorylation levels and inhibition of RHOA activation (Fig. 2f). PROS1 could neither decrease osteoblast matrix mineralization and expression of *Alpl* mRNA nor induce stress fiber formation in the presence of ROCK-inhibitor Y27632 (Fig. 2g–j). These data demonstrate that the PROS1-MERTK axis negatively regulates osteoblastic differentiation and bone formation via the VAV2-RHOA-ROCK pathway.

As RHOA regulates osteoblast motility, our results suggest a role of MERTK in this process. We performed wound healing assays upon the addition of PROS1 with MERTK KO osteoblasts. Here, we observed increased osteoblast migration induced by PROS1 in control conditions, which was abrogated in MERTK KO osteoblasts (SFig. 5a, b). Migrating mesenchymal cells, including osteoblasts, are dependent on a polarized cytoskeletal arrangement and application of cytoskeletal forces to the extracellular matrix via Integrin receptors with focal adhesion formation[34,35]. Analysis of focal adhesion protein vinculin revealed that focal adhesion formation was increased by PROS1.

MERTK KO osteoblasts exhibited decreased vinculin staining intensity and PROS1 could not induce focal adhesions in the absence of MERTK (SFig. 5c, d). Furthermore, PROS1 increased and loss of MERTK decreased the percentage of cells with a leading and a trailing edge, suggesting that MERTK controls polarization of migrating osteoblasts (SFig. 5e). These results indicate that MERTK induces osteoblast migration and focal adhesion formation.

Altogether, our data suggest that MERTK activates the VAV2-RHOA-ROCK pathway in osteoblasts inducing cellular retraction and stress fiber formation, resulting in increased motility, which counteracts osteoblastic bone formation.

## Role of TYRO3 in osteoblasts

Evaluation of *Tyro3−/−OB* mice revealed decreased bone volume with a decreased trabecular number. Trabecular separation was increased, whereas trabecular thickness was not significantly affected (Fig. 3a, b and SFig. 6a–c). Midshaft evaluation of the femur revealed decreased cortical thickness (SFig. 6d). Bone formation rate and osteoblast numbers were decreased, whereas osteoclast numbers were not affected (Fig. 3c–g). These results suggest that TAM receptor TYRO3 promotes osteoblastic bone formation and exerts opposing biological

functions in contrast to MERTK. Correspondingly ex vivo calvarial cell cultures of Tyro3−/−OB mice showed decreased osteoblast matrix mineralization, indicating that TYRO3 directly enhances osteoblast function (Fig. 3h).

We investigated the effects of PROS1 on osteoblast differentiation in the presence and absence of TYRO3. First, we found almost complete knockout of TYRO3 upon the addition of CRE recombinase to Tyro3flox/flox calvarial cells (Fig. 3i). Addition of PROS1 decreased matrix mineralization in TYRO3 KO osteoblast cultures (Fig. 3j). TYRO3 KO osteoblasts showed decreased expression of Alpl and Bglap which was further reduced by treatment with PROS1, suggesting that PROS1-TYRO3 promotes osteoblast differentiation (Fig. 3k, l).

As we observed cytoskeletal regulation of osteoblasts by TAM receptor MERTK, we hypothesized that TYRO3 coming from the same receptor tyrosine kinase family may exert similar functions. Interestingly, we observed opposing effects of TYRO3 vs. MERTK in osteoblast cytoskeletal rearrangement: TYRO3 KO osteoblasts showed enhanced F-actin content with increased stress fiber formation and elevated pRLC activity, which could be further increased by PROS1 (Fig. 3m–p). Concomitantly, migration and focal adhesion formation was increased and PROS1 could further enhance these processes in TYRO3 KO osteoblasts (SFig. 7a–d). In TYRO3 KO osteoblasts, cells containing a leading and a trailing edge were increased and PROS1 could further increase this phenotype (SFig. 7e). These results suggest that the PROS1-TYRO3 axis promotes low F-actin content in osteoblasts with decelerated migration leading to increased osteoblast function.

### Cytoskeletal regulation of osteoblasts by MERTK vs. TYRO3

To validate our findings investigating the biological effects of MERTK and TYRO3 in osteoblasts in the same experiment, we performed siRNA-mediated knockdown of Mertk and Tyro3. Mertk and Tyro3 expression was silenced by 86.2% (SFig. 8a) and 74.4% (SFig. 8b), respectively 3 days after transfection. Confocal microscopy of F-actin immunofluorescence staining revealed that PROS1 increases F-actin intensity, induces stress fiber formation and consistently pRLC intensity in osteoblasts in control conditions. These effects were inhibited by silencing of Mertk (SFig. 9a–e). In contrast, treatment of Tyro3-silenced osteoblasts with PROS1 increased F-actin intensity, induces high-stress fiber formation and increased pRLC intensity, indicating that PROS1-MERTK axis promotes actin polymerization and stress fiber formation, whereas the PROS1-TYRO3 axis inhibits these effects (SFig. 9a–e).

We performed additional wound healing assays upon the addition of PROS1 with Mertk and Tyro3 siRNA-silenced osteoblasts. We observed increased osteoblast migration induced by PROS1 in control conditions, which was abrogated in Mertk-silenced osteoblasts. In contrast, PROS1 treatment in Tyro3-silenced osteoblasts increased osteoblast migration (SFig. 10a, b). These results show that the PROS1-MERTK axis induces osteoblast migration, while the PROS1-TYRO3 axis inhibits migration. Consistently, analysis of focal adhesion protein vinculin revealed that focal adhesion formation was increased by PROS1 in NTsiRNA controls. In Mertk-silenced osteoblasts, PROS1 could not increase vinculin staining intensity, indicating that the PROS1-MERTK axis promotes osteoblast focal adhesion formation. Tyro3-silenced osteoblasts showed increased vinculin staining intensity, which could be further increased by PROS1, indicating that the PROS1-TYRO3 axis inhibits focal adhesion formation (SFig. 10c, d).

Consistently, PROS1 increased the percentage of cells with a leading and a trailing edge in control osteoblasts, whereas in Mertk-silenced osteoblasts, PROS1 treatment could not increase osteoblast polarization. In Tyro3-silenced osteoblasts, cells containing a leading and a trailing edge were increased and PROS1 could further increase this phenotype, indicating that the PROS1-MERTK axis induces and

PROS1-TYRO3 axis inhibits osteoblast polarization (SFig. 11a, b). Cell spreading experiments demonstrated that loss of Mertk accelerates while loss of Tyro3 decelerates osteoblast spreading. PROS1 did not significantly affect spreading capability in our experimental setting (SFig. 11c).

Stimulation with PROS1 led to increased levels of GTP-bound RHOA in control osteoblasts, which was abrogated in Mertk-silenced osteoblasts. In contrast, Tyro3-silenced osteoblasts showed higher PROS1 mediated RHOA activation levels compared to control (SFig. 12).

Altogether our results suggest that MERTK activates the RHOA-ROCK pathway in osteoblasts inducing cellular retraction and stress fiber formation, resulting in increased motility, which counteracts matrix mineralization. In contrast, TYRO3 promotes osteoblast differentiation by antagonizing these effects. Therefore, we hypothesized that MERTK could represent a novel therapeutic target for osteoanabolic pharmacotherapy.

### Effect of pharmacologic MERTK blockade on bone formation in healthy mice

Consequently, we examined the effect of the small molecule R992 on bone formation in healthy mice. R992 is a novel MERTK-selective inhibitor discovered through rational drug design. Figure 4a shows the structure and chemical formula of R992. The drug forms H-bonds with three amino acid residues in the ATP pocket of MERTK (Fig. 4b). R992 is an orally bioavailable, potent, and selective inhibitor displaying low nanomolar cell-based activity against MERTK. In biochemical assays, R992 exhibits seven to eight-fold selectivity over AXL and TYRO3 (Fig. 4c). Furthermore, R992 blocks MERTK phosphorylation in human and murine cells (Fig. 4d, e).

Eight-week-old C57BL/6J mice were treated orally 2x/day with 60 mg/kg R992 continually for 2 weeks. μCT analysis of cancellous bone in R992-treated mice revealed increased bone volume (+59.5%) (Fig.4f, g). Analysis of bone formation revealed an increased bone formation rate (+43.5%) in R992-treated mice (Fig. 4h, i). Consistently, osteoblast cultures showed that R992 could dose-dependently increase bone nodule formation and increase osteoblast differentiation markers Alpl, SP7, and Runx2 (Fig. 4j–l).

### Effect of MERTK blockade on tumor−osteoblast interaction

Skeletal-related events induced by bone-seeking tumors, including multiple myeloma, breast- and lung cancer result in severe pain and fractures, causing substantial morbidity in patients[17,36–38]. The osteolytic disease is caused by an imbalance in bone remodeling, favoring osteoclast-mediated bone resorption over osteoblast-mediated bone formation[36]. As many cancer types express MERTK and TYRO3 as well as their ligand PROS1, we hypothesized that tumor cells may exploit the TAM-receptor-dependent regulation of osteoblasts and inhibit their function via MERTK[39–45]. We utilized the human myeloma cell line U266, the lung cancer cell line H460, and the breast cancer cell line MDA-MB-231. PROS1 expression was confirmed by RT-qPCR in all three cell lines (SFig. 13c). We treated osteoblast cultures with conditioned medium of MDA-MB-231, U266, and H460 cancer cells (referred to as osteoblasttumor cultures as we observed similar results with the three cell lines) in vitro. In osteoblasttumor cultures, osteoblast matrix mineralization was inhibited by these cancer cell lines shown by decreased bone nodule formation. The inhibitory effect on bone formation in vitro could be partly reverted by R992 (SFig. 13a, d). Furthermore, the osteoblast differentiation markers Alpl, Runx2 and Osx were increased in R992-treated osteoblasttumor cultures (SFig. 13e−g). These results suggest that activation of MERTK in osteoblasts may contribute to tumor-induced osteoblast inhibition in myeloma, breast- and lung cancer bone metastasis. Interestingly, migration assays showed that osteoblasttumor cultures exhibited increased motility, which could be reverted by MERTK blockade, indicating that tumor

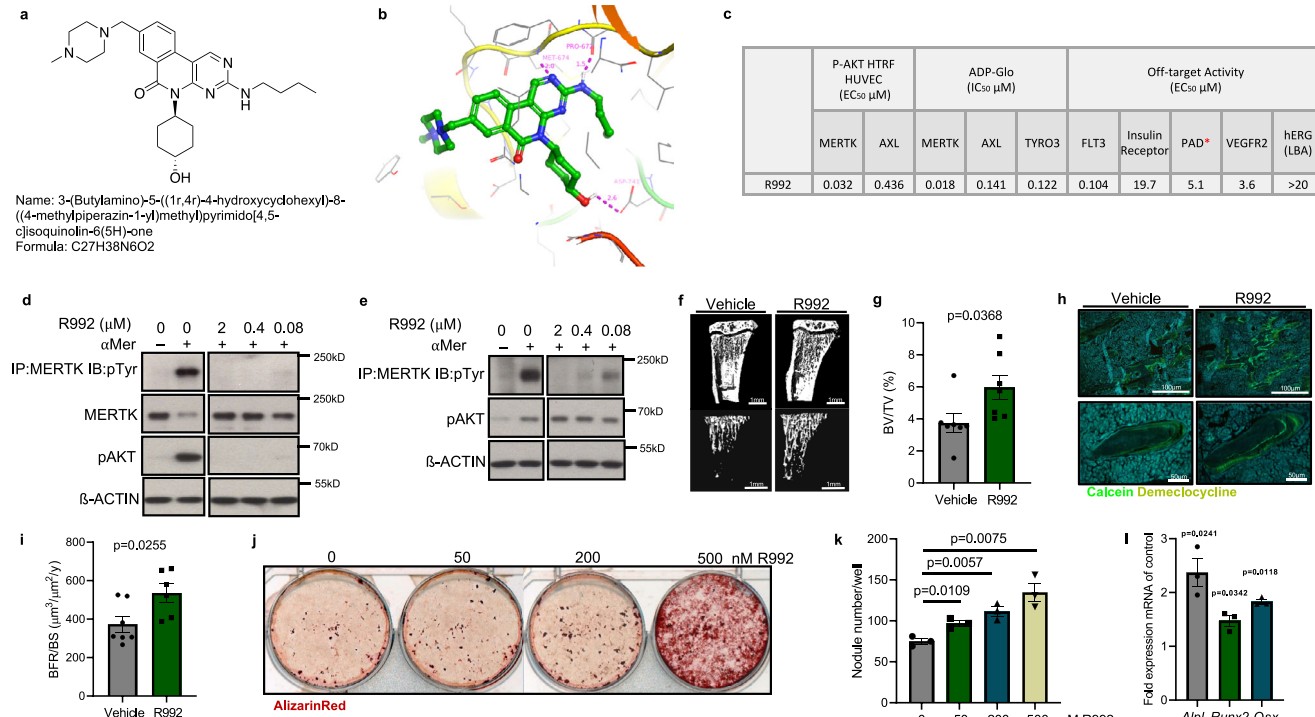

**Fig. 4 | The MERTK-inhibitor R992 induces bone formation in healthy mice.**
**a** Structure and chemical formula of R992: 3-(Butylamino)−5-((1r,4r)−4-hydro-xycyclohexyl)−8-((4-methylpiperazin-1-yl)methyl)pyrimido[4,5-c]isoquinolin-6(5H)-one (C27H38N6O2). **b** Illustration of R992 docked in MERTK and the H-bonds R992 forms with three residues in the ATP pocket (MET-647, PRO-672, and ASP-741). **c** R992 on-target and off-target activity. **d**, **e** Immunoblots of MERTK phosphorylation and AKT signaling in human (**d**) and murine (**e**) cells. **f** µCT of the metaphyseal proximal region of tibias of 10-week-old healthy C57BL/6J mice treated after 2 weeks of treatment with vehicle or R992. **g** Quantification of trabecular bone volume (BV/TV) of the proximal tibia determined by µCT analysis (Vehicle $n = 7$ and R992 $n = 7$). **h**, **i** Representative pictures (**h**) and analysis (**i**) of bone formation rate by Calcein and Demeclocycline double labeling (Vehicle $n = 7$ and R992 $n = 7$). **j**, **k** Representative pictures of Alizarin red staining of calvarial cells cultures treated with different doses of R992 on day 21 (**j**). The number of mineralized nodules was quantified (**k**) ($n = 3$ biological replicates). **l** Analysis of osteoblast differentiation marker *Alpl*, *Runx2*, and *Osx* on day 7 in R992-treated calvarial cell cultures in comparison to control treated cultures ($n = 3$ biological replicates). Data were means ± SEM. Statistical significance was determined by a two-tailed unpaired *t*-test.

cells induce osteoblast motility which counteracts osteoblast bone-forming activity (SFig. 13b, h–j). Analysis of F-actin cytoskeleton by confocal microscopy revealed that cancer cells indeed promote complex conformational changes in osteoblast morphology with increased F-actin content and stress fiber formation (SFig. 14a, b). These changes were accompanied by the formation of spindle cells, a sign of immaturity in osteoblast lineage[46] (SFig. 14a). Treatment with R992 of osteoblast[tumor] cultures could decrease F-actin content, stress fiber formation, and promoted osteoblast spreading (SFig. 14a–d). These data suggest that tumor cells induce a cytoskeletal reorganization in osteoblasts leading to retraction via MERTK, which inhibits osteoblast differentiation and bone-forming capacity and induces osteoblast motility. Therefore, MERTK could represent a target to reduce cancer-induced osteoblast inhibition during osteolytic bone disease.

### MERTK blockade in multiple myeloma tumor progression and osteolytic bone disease

In the next step, we assessed if MERTK blockade could reduce cancer-induced bone loss and/or tumor progression in multiple myeloma. After intraosseus injection of the U266 cell line, myeloma tumor burden was monitored by assessing lambda light chain concentration in the blood. In concordance with our previously published results that genetic ablation of *Mertk* in myeloma cells inhibits tumor progression in vivo[24] we could observe that pharmacologic MERTK blockade by R992 led to reduced tumor burden and bone marrow infiltration after 8 weeks in the U266 myeloma mouse model (Fig. 5a, b) (Supplementary Methods). We confirmed the efficacy of R992 in inhibiting tumor progression in myeloma in a second myeloma

mouse model using a human RPMI8226 cell line (SFig. 15a, b). For studying myeloma bone disease, we utilized the U266 mouse model and performed µCT and histology. Analysis of metaphyseal cancellous bone by µCT showed reduced bone volume induced by U266 cells in comparison to age-matched healthy controls after 5 weeks, indicating induction of osteolytic bone disease. Treatment of myeloma-bearing mice with R992 led to increased bone volume (Fig. 5c, d and SFig. 16a–d). We found increased osteoblast numbers on the bone surface and elevated levels of the serum bone formation marker procollagen type 1 N propeptide (P1NP), associated with the preserved bone volume induced by R992 in the myeloma mouse model in concordance with a stimulatory effect on bone formation (Fig. 5e–g). Analyses of osteoclasts by histomorphometry and determination of serum TRAP5b, showed that osteoclast numbers were not significantly affected by R992 treatment (Fig. 5h, i). The function of MERTK in osteoclasts, which derive from bone marrow macrophages, has not been investigated so far, but in line with the crucial function of MERTK for directional migration in macrophages[47], we observed decreased numbers of TRAP+ mononuclear cells in the bone marrow suggesting that MERTK blockade might affect sufficient osteoclast precursor cell recruitment to bone resorption sites (Fig. 5e, j). Altogether, our results suggest that R992 antagonizes myeloma-induced bone loss mainly by promoting osteoblastic bone formation. Consistently, we could observe signs of osteoblastic bone repair in disrupted cortical bone in R992-treated mice, which underlines this hypothesis (Fig. 5e). Evaluation of survival revealed increased overall survival in R992-treated mice in the U266 myeloma mouse model (Fig. 5k).

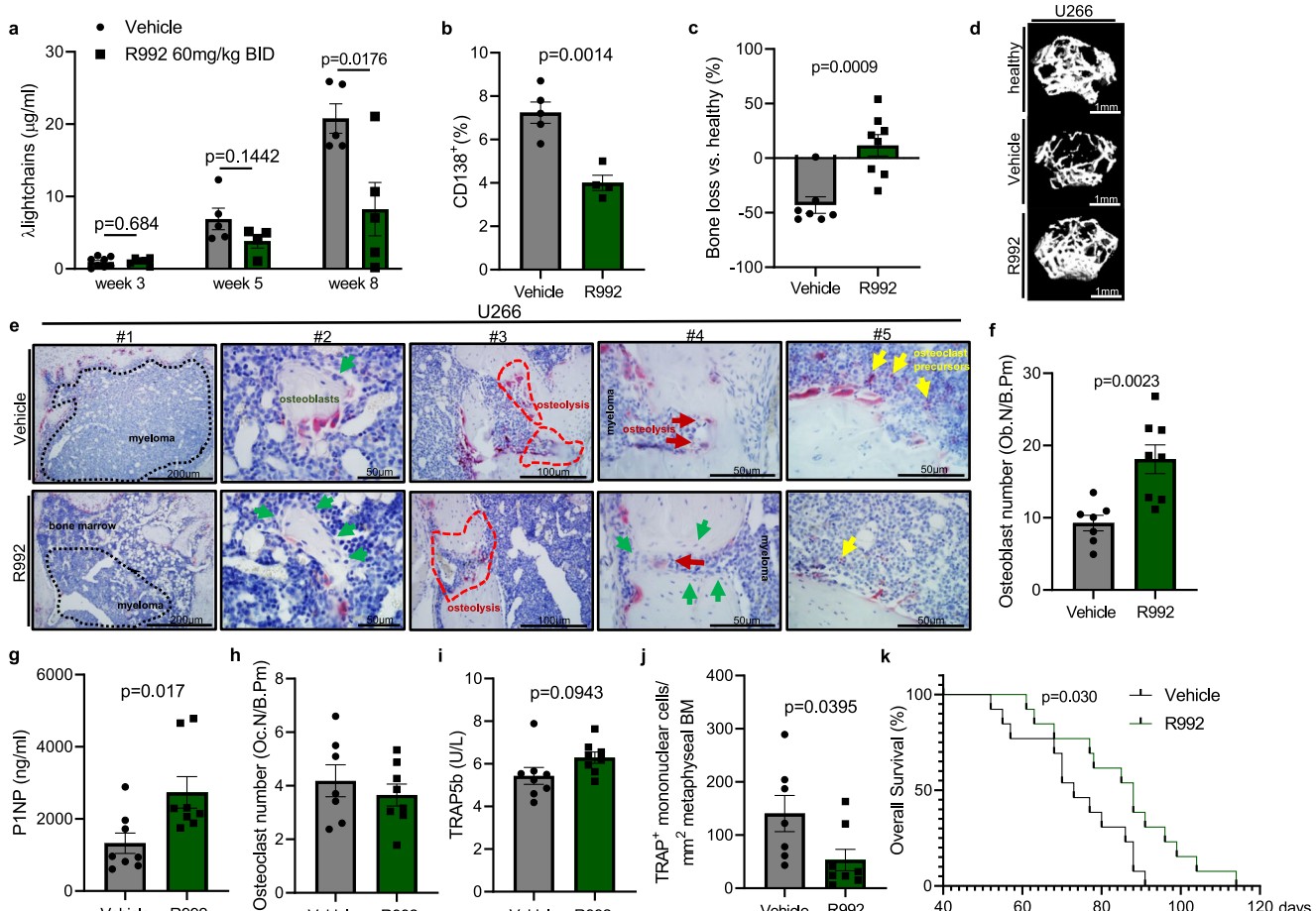

**Fig. 5 | Pharmacologic MERTK blockade inhibits multiple myeloma tumor progression and bone disease. a–k** After intraosseous femoral injection of U266 myeloma cell line, NSG mice were treated with vehicle or 60 mg/kg R992 BID. **a** Assessment of Igλ light chain paraprotein in the peripheral blood by ELISA (week 3 $n = 8/6$, week 5 $n = 5/4$, and week 8 $n = 5/5$). **b** Bone marrow CD138[+] myeloma plasma cell infiltration in the contralateral leg by FACS ($n = 5/4$). **c** µCT analysis of bone loss by calculating the change in bone volume (BV/TV) in relation to healthy control mice ($n = 7/8$). **d** 3D reconstructions of microcomputed tomography (µCT) recordings of the metaphyseal proximal region of the femur of healthy and myeloma-bearing mice. **e** Representative pictures of TRAP Hematoxylin staining (osteoclasts stained in red) showing myeloma infiltration in the bone marrow (#1), osteoblasts visible as cuboidal or polygonal mononuclear cells on the endosteal bone surface (arrows in green) (#2), osteolysis (#3), histologically detectable signs

of bone repair in osteolysis zones induced by R992 (arrows in green pointing to osteoblasts, arrows in red pointing to osteoclasts) (#4), and detection of TRAP-positive mononuclear cells indicating osteoclast precursor cell recruitment (#5) (arrows in yellow) ($n = 5$). **f** Histomorphometric analysis of osteoblast number per bone perimeter ($n = 7/8$). **g** Determination of serum bone formation marker P1NP by ELISA ($n = 8/8$). **h** Histomorphometric analysis of osteoclast number per bone perimeter. **i** Determination of serum osteoclast marker TRAP5b by ELISA ($n = 8/8$). **j** Histomorphometric analysis of the number of TRAP[+] mononuclear cells in the bone marrow ($n = 7/8$). **k** Kaplan–Meier curve showing overall survival ($n = 13/13$, $p = 0.030$ Mantel–Cox test). Median survival was 73 vs. 88 days. Data were means ± SEM. Statistical significance was determined by a two-tailed unpaired $t$-test unless otherwise stated.

## Effect of pharmacologic MERTK blockade on bone metastasis

Further studies were conducted to assess the efficacy of R992 in metastasized bone-seeking solid tumors. Therefore, we injected luciferase-transduced H460 lung cancer and MDA-MB-231 breast cancer cells intracardially in NSG mice to induce metastatic spread.

Bioluminescence imaging revealed that intracardiac injection of luciferase transduced H460 and MDA-MB-231 cells resulted in rapid induction of bone metastasis as well as metastasis in non-skeletal tissues. MERTK blockade led to an overall reduction of metastasis burden as well as tumor growth in the brain, spine, liver, and hind limbs in the H460 lung cancer model, whereas in the MDA-MB-231 breast cancer model tumor growth was not statistically significantly affected (Fig. 6a–c and SFig. 17a, b). µCT analysis of metaphyseal tibia in the H460 and MDA-MB-231 mouse models showed reduced bone volume in comparison to age-matched controls, indicating induction of osteolytic bone disease. Treatment of tumor-bearing mice with R992 led to increased bone volume (Fig. 6d, e and SFigs. 18a–d, 19a–d).

Histomorphometric analysis of osteoblasts revealed consistently increased numbers in both tumor models (Fig. 6f, g and SFig. 20a, b). Levels of serum bone formation marker procollagen type 1 N propeptide (P1NP) were augmented in both mouse models upon treatment, indicating increased bone formation induced by R992 (Fig. 6h). Evaluation of osteoclasts showed that in the MDA-MB-231 mouse model their number was not affected, whereas osteoclast numbers in the H460 mouse model were slightly decreased (Fig. 6i). Serum TRAP5b levels were increased in R992-treated MDA-MB-231 mice but showed no significant changes in the H460 model (Fig. 6j). Similar to our observations in the myeloma model we observed decreased TRAP-positive mononuclear cells in the bone marrow in the H460 mouse model and intratumorally in the MDA-MB-231 model (Fig. 6k, l). These data indicate that while we cannot exclude effects on osteoclasts, the effects of MERTK blockade on osteoblast phenotype prevail in vivo. Evaluation of survival revealed increased overall survival in R992-treated mice in the lung- and breast cancer model (Fig. 6m, n).

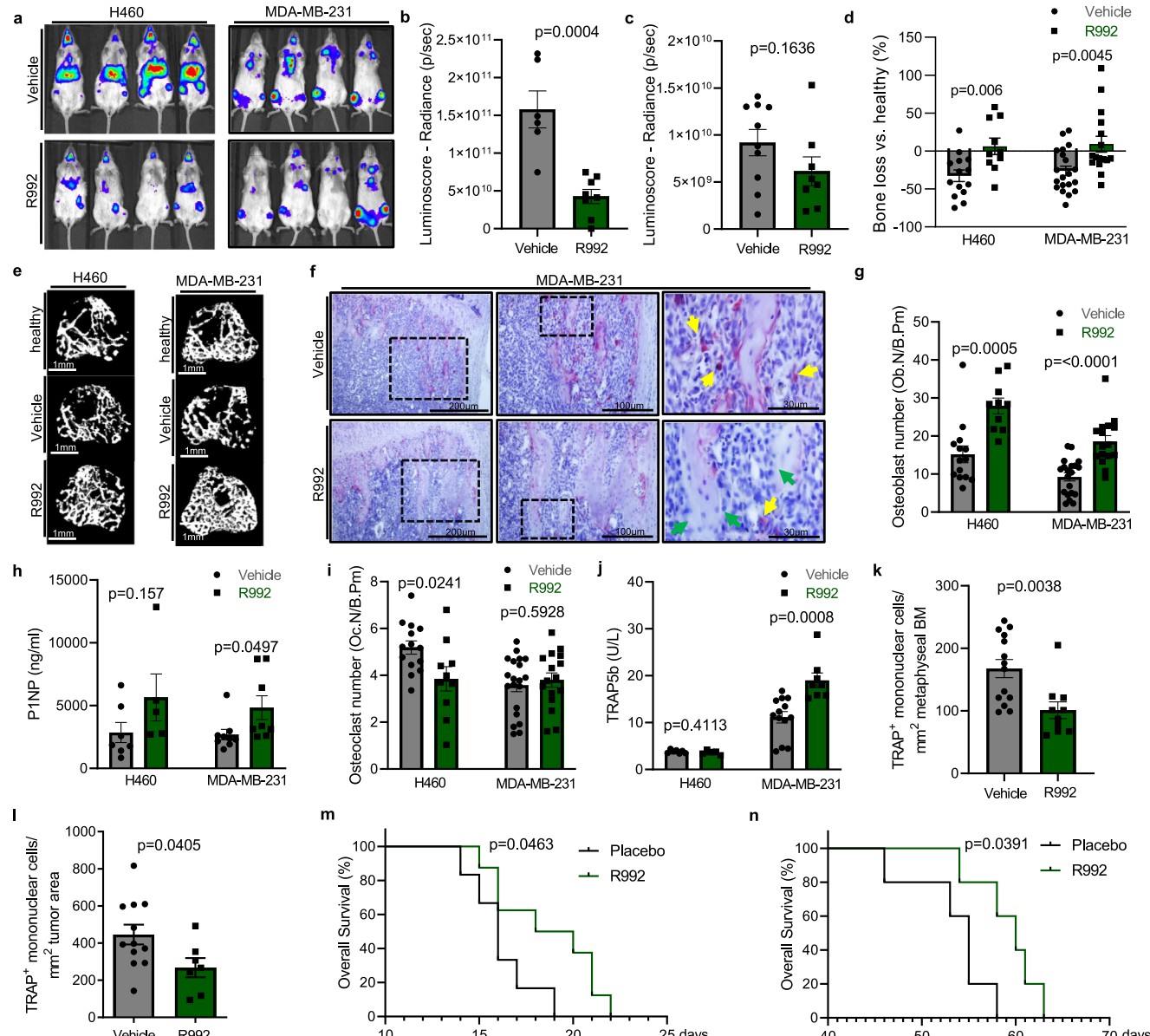

**Fig. 6 | Pharmacologic MERTK blockade inhibits tumor progression and cancer-induced bone loss in breast- and lung cancer bone metastasis. a–n** After intracardiac injection of luciferase⁺ H460 and MDA-MB-231 cells, NSG mice were treated with vehicle or 60 mg/kg R992 BID. **a** Representative images from bioluminescence imaging indicating the amount of tumor load. **b**, **c** Luminoscore (whole-body radiance intensity) in the H460 (**b**) and MDA-MB-231 (**c**) mouse model (H460 model: Vehicle $n = 6$ and R992 $n = 8$; MDA-MB-231 model: Vehicle $n = 10$ and R992 $n = 8$). **d** μCT analysis of bone loss by calculating the change in bone volume (BV/TV) of tumor-bearing mice in relation to healthy age-matched control mice (H460 model: healthy controls $n = 5$, vehicle $n = 14$, and R992 $n = 10$; MDA-MB-231 model: healthy controls $n = 8$, vehicle $n = 20$, and R992 $n = 16$). **e** 3D reconstructions of microcomputed tomography (μCT) recordings of the metaphyseal proximal region of the tibia of healthy and tumor-bearing mice treated with vehicle or R992. **f** Representative pictures of TRAP/Hematoxylin staining of bone metastasis of MDA-MB-231 injected mice. Osteoclasts are stained in red, green arrows pointing to osteoblasts, and yellow arrows pointing to intratumoral TRAP-positive

mononuclear cells. **g** Histomorphometric analysis of osteoblast number (H460: vehicle $n = 14$ and $n = 10$; MDA-MB-231: vehicle $n = 20$ and R992 $n = 16$). **h** Determination of serum bone formation marker P1NP by ELISA (H460: vehicle $n = 7$ and R992 $n = 5$; MDA-MB-231: vehicle $n = 9$ and R992 $n = 8$). **i** Histomorphometric analysis of osteoclast number per bone perimeter (H460: vehicle $n = 14$ and R992 $n = 10$; MDA-MB-231: vehicle $n = 20$ and R992 $n = 16$). **j** Determination of serum osteoclast marker TRAP5b by ELISA (H460: vehicle $n = 7$ and R992 $n = 5$; MDA-MB-231: vehicle $n = 10$ and R992 $n = 8$). **k**, **l** Histomorphometric analysis of the number of TRAP⁺ mononuclear cells in the bone marrow in the H460 model (vehicle $n = 14$ and R992 $n = 10$) (**k**) and in the MDA-MB-231 model (vehicle $n = 12$ and R992 $n = 7$) (**l**). **m**, **n** Kaplan–Meier curves of overall survival in the H460 (**m**) and MDA-MB-231 (**n**) model (H460: vehicle $n = 6$ and R992 $n = 8$; $p = 0.0463$ Mantel–Cox test; Median survival 16 vs. 19 days) (MDA-MB-231: vehicle $n = 5$ and R992 $n = 5$; $p = 0.0391$ Mantel–Cox test; Median survival 55 vs. 60 days). Data were mean ± SEM. Statistical significance was determined by a two-tailed unpaired $t$-test unless otherwise stated.

## Effect of *Mertk* deficiency in osteoblasts on metastasis-induced bone loss

To validate, that MERTK in osteoblasts plays a role in tumor–osteoblast interaction, we treated MERTK KO osteoblast cultures again with a conditioned medium of MDA-MB-231, U266, and H460 cancer cells. In osteoblast^tumor cultures, the inhibitory effect on the bone formation of

the human cancer cell lines was partly reversed in MERTK KO osteoblast cultures (SFig. 21a). Furthermore, the osteoblast differentiation marker *Alpl, Runx2*, and *Osx* were increased in MERTK KO osteoblast^tumor cultures (SFig. 21b–d). These results suggest that activation of MERTK in osteoblasts may contribute to tumor-induced osteoblast inhibition in myeloma, breast-, and lung cancer bone metastasis. Migration assays

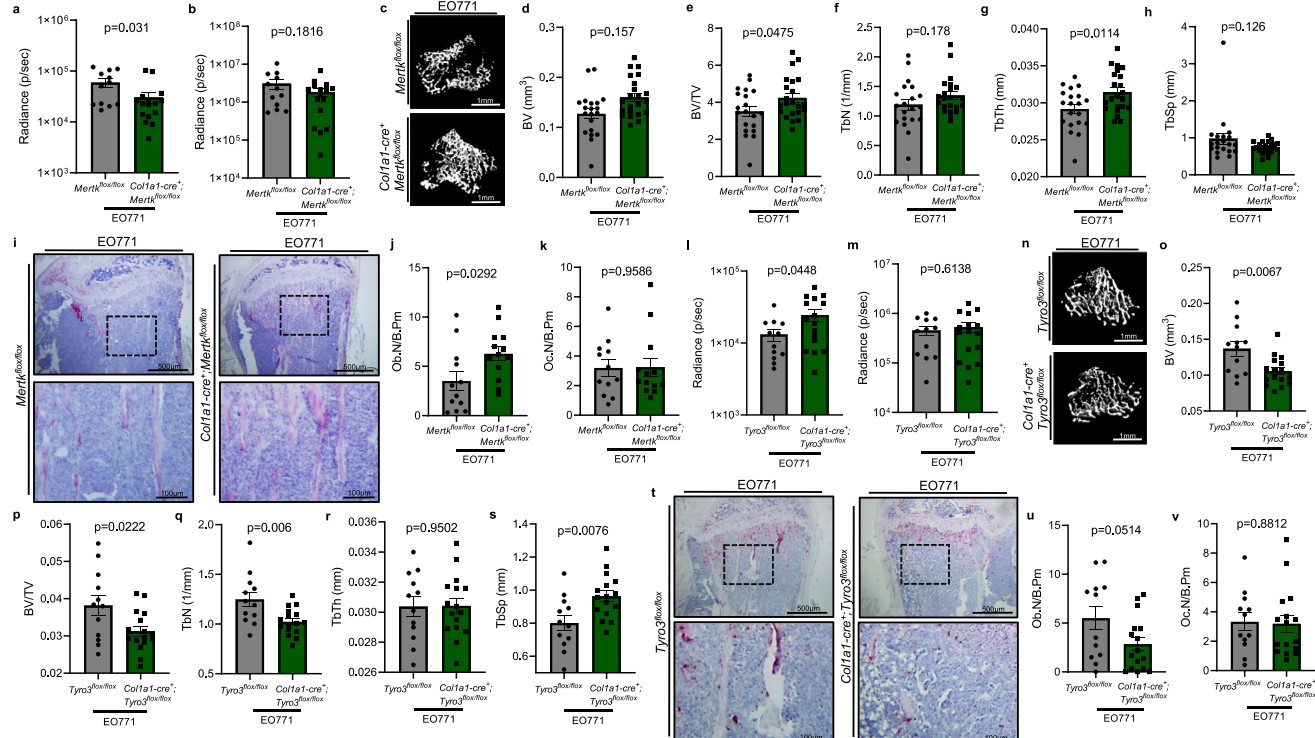

**Fig. 7 | *Mertk* deficient mice display reduced bone loss in a syngeneic breast cancer bone metastasis mouse model. a–k** Luciferase⁺ EO771 breast cancer cells were injected intracardially in *Mertk^{flox/flox}* and *Col1a1-cre⁺;Mertk^{flox/flox}* mice. **a, b** Analysis of tumor load in the tibia by Bioluminescence Imaging after 3 (*n* = 12/16) (**a**) and 7 (*n* = 12/16) days. **c–h** µCT 3D reconstructions (**c**) and analysis of BV (*n* = 20/22) (**d**), BV/TV (*n* = 20/22) (**e**), Tb.N (*n* = 20/22) (**f**), Tb.Th (*n* = 20/22) (**g**), and Tb.Sp (*n* = 20/22) (**h**) of trabecular bone of metaphyseal proximal region of the tibia. **i** Representative pictures of TRAP/Hematoxylin staining of bone metastasis of EO771 injected mice. **j, k** Histomorphometric analysis of osteoblast number (Ob.N/B.Pm) (*n* = 12/13) (**j**) and osteoclast number (Oc.N/B.Pm) (*n* = 12/13) (**k**).

**l–v** Luciferase⁺ EO771 breast cancer cells were injected intracardially in *Col1a1-cre⁺; Tyro3^{flox/flox}* mice. **l, m** Analysis of tumor load in the tibia by Bioluminescence Imaging after 3 (*n* = 12/16) (**l**) and 7 (*n* = 12/16) (**m**) days. **n–s** µCT 3D reconstructions (**n**) and analysis of BV (*n* = 12/16) (**o**), BV/TV (*n* = 12/16) (**p**), Tb.N (*n* = 12/16) (**q**), Tb.Th (*n* = 12/16) (**r**), and Tb.Sp (*n* = 12/16) (**s**) of trabecular bone of metaphyseal proximal region of the tibia. **t** Representative pictures of TRAP/Hematoxylin staining of bone metastasis of EO771 injected mice. **u, v** Histomorphometric analysis of osteoblast number (Ob.N/B.Pm) (*n* = 12/16) (**u**) and osteoclast number (Oc.N/B.Pm) (*n* = 12/16) (**v**). Data were means ± SEMs. Statistical significance was determined by a two-tailed unpaired *t*-test unless otherwise stated.

showed that osteoblast^{tumor} cultures exhibited increased motility, which could be reverted by MERTK KO, indicating that tumor cells induce osteoblast motility which counteracts osteoblast bone-forming activity (SFig. 21e–h). Analysis of the F-actin cytoskeleton by confocal microscopy revealed that MERTK KO osteoblast^{tumor} cultures exhibited markedly decreased F-actin content and stress fiber formation (SFig. 21i–k). IF staining of pRLC showed increased staining intensity induced by cancer cells. In turn, MERTK KO osteoblast^{tumor} cultures exhibited decreased RLC activity (SFig. 21i, l). These data suggest that tumor cells induce a cytoskeletal reorganization in osteoblasts leading to retraction via MERTK, which inhibits osteoblast differentiation and bone-forming capacity and induces osteoblast motility.

To further provide evidence that MERTK in osteoblasts represents a novel target to increase bone volume in bone-seeking tumors, we used the syngeneic EO771 breast cancer bone metastasis model and injected luciferase transduced EO771 cells into *Mertk^{flox/flox}* and *Col1a1-cre⁺;Mertk^{flox/flox}* C57BL/6J mice.

The metastatic spread was monitored by bioluminescence imaging and µCT imaging was performed after 12 days. Tumor load in hind limbs was slightly decreased in *Col1a1-cre⁺;Mertk^{flox/flox}* mice 3 days after injection, but was not significantly changed after 1 week (Fig. 7a, b). Analysis of metaphyseal cancellous bone by µCT showed higher bone volume and trabecular thickness in EO771 tumor-bearing *Col1a1-cre⁺;Mertk^{flox/flox}* mice in comparison to *Mertk^{flox/flox}* mice after 12 days (Fig. 7c–h). Histomorphometry of TRAP/Hematoxylin staining indicates increased osteoblast numbers in EO771 tumor-bearing *Col1a1-cre⁺;Mertk^{flox/flox}* mice, whereas osteoclast numbers were not changed (Fig. 7i–k).

## Effect of *Tyro3* deficiency in osteoblasts on metastasis-induced bone loss

Conversely, injection of EO771 cells into *Tyro3^{flox/flox}* and *Col1a1-cre⁺; Tyro3^{flox/flox}* C57BL/6J mice led to slightly increased hind limb tumor load, whereas tumor load in the later timepoint was not changed (Fig. 7l, m). Analysis of metaphyseal cancellous bone by µCT showed decreased bone volume and trabecular number in EO771 tumor-bearing *Col1a1-cre⁺;Tyro3^{flox/flox}* mice in comparison to *Tyro3^{flox/flox}* mice after 12 days (Fig. 7n–s). Histomorphometry of TRAP/Hematoxylin staining revealed decreased osteoblast numbers in EO771 tumor-bearing *Col1a1-cre⁺;Tyr-o3^{flox/flox}* mice, whereas osteoclast numbers were not changed (Fig. 7t–v).

These data confirm that activation of MERTK and not TYRO3 on osteoblasts promotes breast cancer-induced bone loss. Therefore, MERTK represents a novel target whose inhibition counteracts cancer-induced osteopenia through stimulation of osteoblast function, whereas TYRO3 represents a bone protective factor.

## Discussion

Altogether we could demonstrate by using osteoblast-specific in vivo loss of function mouse models that TAM receptors MERTK and TYRO3 control osteoblast-mediated bone formation, with MERTK inactivation increasing and TYRO3 inhibition decreasing bone mass. Utilization of other osteoblast lineage cre lines acting at different stages of osteoblast development could provide stronger phenotypes and will be subject to further studies.

On a molecular level, we showed that the TAM-receptor ligand PROS1 activates the MERTK-VAV2-RHOA-ROCK axis favoring

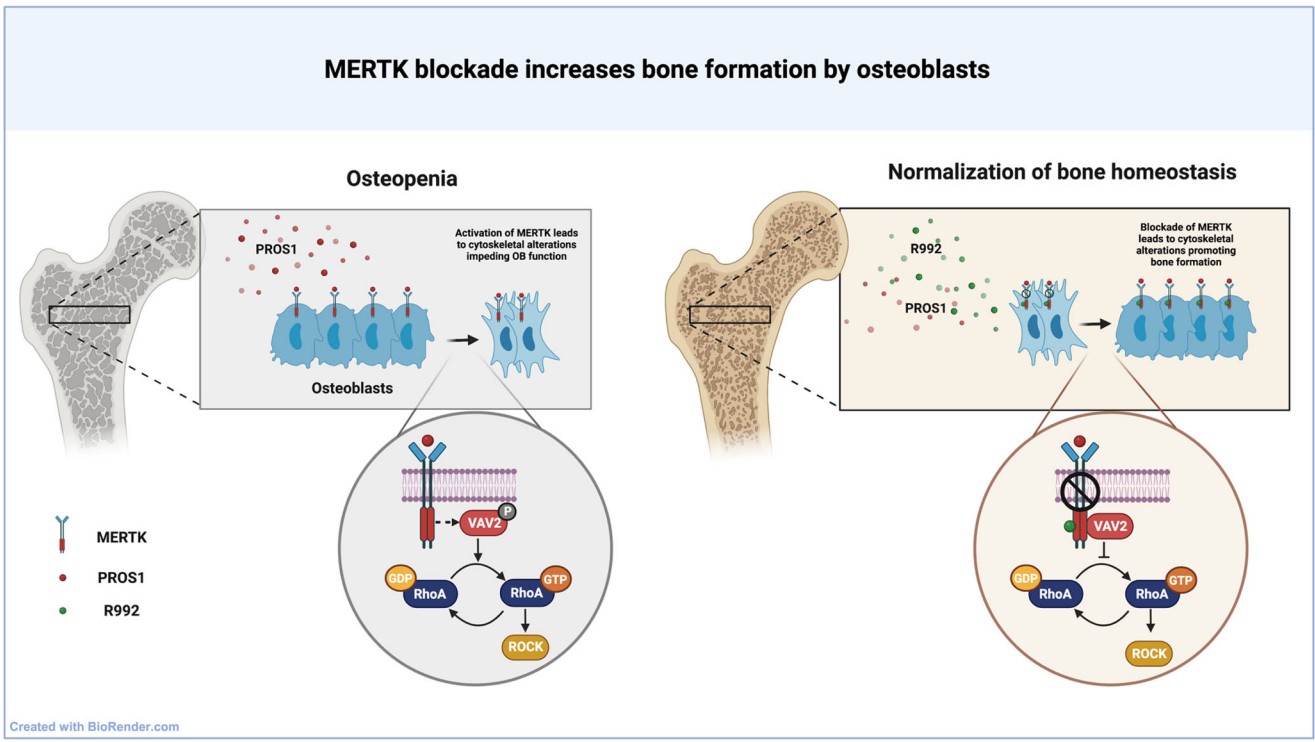

**Fig. 8 | Schematic outline of the mechanism showing how the MERTK-PROS1 axis exerts osteopenia via the VAV2-RHOA pathway.** Inhibition of this axis led to the normalization of bone homeostasis by increasing osteoblast function.

osteoblast motility over differentiation and bone formation. The regulation of osteoblasts by MERTK has a profound impact on cell morphology and the F-actin cytoskeleton: MERTK promotes activation of the F-actin cytoskeleton with retraction and polarization, which induces migration. Interestingly, we observed antagonistic phenotypic effects of TYRO3 vs. MERTK, where TYRO3 induces a cell morphology displaying low F-actin content and decreased migration. To our knowledge, these potential converse effects of MERTK and TYRO3 in the same cell type have not yet been described. Future studies are needed, to investigate if these receptors exert a direct functional antagonism and if a potential homeostatic cytoskeletal control by MERTK and TYRO3 is applicable to further mammalian cell types.

MERTK and TYRO3 are activated by the same signaling molecules. So how is their relative activation in osteoblasts controlled in the bone? The ligand PROS1 is mostly known as a cofactor of protein C, together with which it exerts important anticoagulant activity. PROS1 is present in considerably high plasma levels of about 300 nM[48]. Due to the anticoagulant function *Pros1* KO mice are embryonic lethal, which makes the general absence of PROS1 difficult to study in vivo[49]. Nevertheless, because PROS1 can bind to MERTK as well as TYRO3, the question arises of which receptor is activated preferentially in osteoblasts by PROS1 in vivo. Our in vitro results suggest that PROS1 negatively regulates osteoblast differentiation despite the presence of TYRO3. These data indicate that the PROS1-MERTK axis is dominant over the PROS1-TYRO3 axis in osteoblast differentiation. The dominance of the PROS1-MERTK axis could be explained by increasing levels of MERTK during osteoblast differentiation, while TYRO3 levels remain constant and are considerably lower in comparison to MERTK levels. Further work is warranted to dissect the regulation of the MERTK vs. TYRO3 pathway in osteoblasts in health and disease conditions, including osteoporosis.

Until today antiresorptive therapies, such as bisphosphonates and the receptor activator of nuclear factor-κB ligand (RANKL) antibody denosumab, are dominating the armamentarium for the treatment of osteoporosis[36]. Their mechanism of action is to slow down bone remodeling by reducing the number and activity of osteoclasts, thus abating the decline in bone volume and microstructural damage. Nevertheless, antiresorptive agents cannot repair the destructed bone or cure the disease[50]. Besides the emerging anti-sclerostin antibodies, which promote bone formation by increasing canonical WNT−β-catenin signaling in osteoblasts, therapeutic options for osteoporosis or cancer-induced bone disease are lacking[50], highlighting the unmet medical need for osteoanabolic agents.

We could demonstrate that MERTK blockade by the small molecule R992 increases bone volume and bone formation in mice, suggesting that the blockade of MERTK represents a novel osteoanabolic treatment strategy (Fig. 8).

It should be noted that small molecule inhibitors frequently inhibit other receptor tyrosine kinases. In the case of R992, the most prominent off-targets are TYRO3 and AXL (Fig. 4). Pharmacokinetics of R992 in C57BL/6J mice were measured previously. It was observed that treatment of mice with 60 mg/kg BID led to trough concentrations of 260 ng/ml (~50 nM) which increased up to 2020 ng/ml (~400 nM) 0.5h-1h after oral gavage. In our cell culture experiments, we observed dose-dependently decreased MERTK phosphorylation with 80 and 400 nM R992, which corresponds to the aforementioned dose levels in mice reached with 60 mg/kg BID[51].

The conclusion that the observed effects of R992 in vivo are predominantly mediated via MERTK are additionally strengthened by the findings that osteoblast-specific deletion of *Mertk* phenocopies the effects of R992 in the tumor context leading to increased bone mass, while deletion of *Tyro3* has the opposite effect. Thus, if R992 would lead to a relevant inhibition of TYRO3 in addition to blocking MERTK, both effects would outweigh each other, which is not the case.

The role of AXL in osteoblasts has not been studied yet. Other more specific pharmacologic approaches to block MERTK signaling, including a MERTK-specific function-blocking antibody are interesting and subject to future investigation.

The combination of the tumor-promoting- and osteoblast-inhibitory functions of MERTK makes this therapeutic strategy

especially attractive in bone-seeking tumors because tumor progression and bone loss is treated simultaneously. Consistently in three different cancer models, we could observe increased bone volume and P1NP levels. Importantly, the osteoanabolic effects occur independently of the amount of infiltrating tumor cells, as we observed increased bone mass, while tumor load was not changed in the MDA-MB-231 breast cancer model. This treatment could be of special interest in myeloma patients who suffer from sustained bone fragility and osteolysis even in complete remission of the malignant plasma cell clone[52]. This myeloma-induced bone disease could be targeted by inhibiting MERTK with R992 or other agents. As we used immuno-compromised NSG mice in our cancer models, an immune-related effect leading to increased bone volume by MERTK blockade is unlikely. The important function of MERTK in macrophages could point to an additional role in bone homeostasis by supporting osteoclast differentiation and function, which may also contribute to the observed osteoanabolic effects of MERTK blockade. Future studies about the function of MERTK in osteoclasts are needed to deepen the knowledge of how MERTK controls bone remodeling.

Altogether, MERTK holds promise as a strategy to treat bone and joint diseases, including cancer-induced osteolytic bone disease and potentially also osteoporosis or rheumatoid arthritis.

## Methods

### Animal experiments

For osteoblast-specific knockouts, Col1a1-2.3-kb-Cre mice with C57BL/6J background were crossed with $Mertk^{flox/flox}$ and $Tyro3^{flox/flox}$ mice with C57BL/6J background (SFig. 1)[7]. The presence of the Cre transgene and the homozygous status of the floxed genomic region was determined in tail biopsies by specific PCR protocols. Eight-week-old females were used for bone analysis in healthy mice.

For xenograft mouse models, 6–8-week-old female NSG (NOD.C-PrkdcscidIl2rgtm1Wjl/SzJ) mice were used.

U266 wild-type cells ($2 \times 10^6$) were injected into the femur of 6–8 weeks old female NSG mice. Myeloma tumor load was monitored by assessing λ light chain concentration in the blood plasma using ELISA according to the manufacturer's instructions (Human Lambda ELISA KIT; Bethyl Laboratories Inc). For the survival study animals were considered end-stage and euthanized by signs of beginning hind limb paralysis, weight loss >20% or extended weakness and lethargy. U266 myeloma cells induce osteolytic bone disease with severe cortical lesions already 6 weeks after injection[21]. For the myeloma bone disease study, we injected U266 MM cells and led the cells to engraft for 10 days. Afterward, R992 treatment was started (60 mg/kg BID). Mice were euthanized after 5 weeks to assess myeloma bone disease by uCT analysis and histology in the injected femur.

RPMI8226 cells were lentivirally transduced with LeGO-Venus (LeGO-V) vector for stable expression of the Venus fluorescence gene. $2 \times 10^6$ cells were injected into the femur of 6–8 weeks old female NSG mice. Myeloma tumor load was monitored by assessing λ light chain concentration in the blood plasma using ELISA. RPMI8226 cells in the injected leg were quantified by assessing Venus fluorescence using FACS Fortessa (BD).

$1.0 \times 10^5$ MDA-MB-231-Luc, H460-Luc, or EO771-Luc cells were injected into the left cardiac ventricles of 6–8 weeks old female mice. For MDA-MB-231-Luc and H460-Luc, NSG mice were used. For EO771-Luc $Tyro3^{flox/flox}$ and Col1a1-cre[+];$Tyro3^{flox/flox}$ C57BL/6J mice or $Mertk^{flox/flox}$ and Col1a1-cre[+];$Mertk^{flox/flox}$ C57BL/6J mice were used. We started treatment with R992 (60 mg/kg BID) after 2 days (H460-Luc) or 7 days (MDA-MB-231-Luc) due to different growth kinetics. In vivo whole-body bioluminescence imaging was performed by intraperitoneal injection of Luciferin and subsequent image acquisition with the IVIS 200 imaging system. Regions of interest (ROIs) of specific anatomic regions were determined and quantified. Luminoscore was calculated by the addition of front and back whole-body radiance intensity. The

photon emission transmitted from the ROIs was quantified in photons/sec. On day 9 (H460-Luc), day 21 (MDA-MB-231-Luc), or day 12 (EO771-Luc) after tumor injection, mice were euthanized to study bone metastasis and osteolytic bone disease by μCT analysis and histology.

For in vivo experiments, R992 was dissolved according to the manufacturer's instructions and treated BID per oral gavage. The dissolvent without the addition of R992 was used as a control treatment (Vehicle treatment).

All animal experiments were carried out in concordance with the institutional guidelines for the welfare of animals in experimental neoplasia and were approved by the local licensing authority (Behörde für Soziales, Gesundheit, Familie, Verbraucherschutz; Amt für Gesundheit und Verbraucherschutz, project number G65/17, N24/19, N30/19, and N119/21). Housing, breeding, and experiments were performed under a 12 h light–12 h dark cycle and standard laboratory conditions ($22 \pm 1\,°C$, 55% humidity, food, and water ad libitum, and 150–400 lx light intensity during the light phase).

### Cell culture

U266 (obtained from DSMZ, no. ACC 9), RPMI8226 (obtained from DSMZ, no. ACC 402), H460 (the gift from Klaus Pantel), and MDA-MB-231 (the gift from Klaus Pantel) cells were cultured in RPMI1640 medium supplemented with 10% fetal bovine serum (FBS) and 1% penicillin/streptomycin (P/S). EO771 (the gift from Massimiliano Mazzone) cells were cultured in DMEM medium supplemented with 10% fetal bovine serum (FBS) and 1% penicillin/streptomycin (P/S). Human cell lines were frequently authenticated using the Multiplex human Cell line Authentication Test (MCA) by Multiplexion. The last authentication was performed for MDA-MB-231 and U266 in 2021 and for H460 and RPMI822 in 2020 after the indicated experiments.

The human breast cancer cell line MDA-MB-231-Luc, human non-small cell lung cancer (NSCLC) cell line H460-Luc and the murine breast cancer cell line EO771-Luc were generated by lentivirally transducing them with the firefly luciferase gene. The third generation HIV-1 derived lentiviral vector LeGO-iG2-Puro[+]-Luc2 expresses the Luc2 variant of firefly luciferase (cloned from AddGene Plasmid 24337) under the control of an SFFV-promoter, linked by an IRES to a second open reading frame consisting of eGFP, a 2A-peptide and the puromycin resistance PAC[53,54]. After lentiviral transduction, luciferase[+] cells were selected by Puromycin treatment (1 mg/ml) and purity >99% was confirmed by FACS analysis. The sufficient luciferase signal of the cells was validated in vitro by several dilution series, luciferin treatment, and image acquisition with IVIS 200 imaging system (PerkinElmer).

Cells from calvaria of 2–3 days old C57BL/6J mice were extracted by four consecutive enzymatic digestions with 1 mg/ml Collagenase and 2 mg/ml Dispase ll and seeded out on 10 cm Petri dishes with standard aMEM Medium (α-MEM supplemented with 10% FBS, 1% Penicillin/Streptomycin, and 1% L-Glutamine). When cells were confluent, they were seeded out based on the indicated experimental procedure with a standard α-MEM medium. After 24 h medium was exchanged to osteogenic medium (50 μg/ml ascorbic acid, 10 mM beta-glycerophosphate, and 10 nM Dexamethasone) and replaced every 3 days. For analysis of alkaline phosphatase activity, 0.1 M Tris–HCl buffer (pH. 8.4) with 0.25 mg/ml Naphthol AS-MX phosphate disodium salt (Sigma) in N,N dimethylformamide and 0.5 mg/ml fast blue RR (Sigma) was used. Analysis of osteoblast differentiation markers Alpl, Runx2, and SP7 (Osterix) was performed by RT-qPCR using Taqman probes (Thermo Fisher Scientific). Bone nodule formation was quantified on day 21 by using 0.04 mg/ml Alizarin Red staining solution (pH = 4.2) (Sigma).

We utilized Cre Recombinase Gesicles (TakaraBio) to deliver recombinant Cre recombinase to osteoblasts according to the manufacturer's instructions and treated ex vivo calvarial cell cultures of $Mertk^{flox/flox}$ and $Tyro3^{flox/flox}$ mice. Efficient knockout was confirmed by SDS-PAGE.

Calvarial cells were transfected using Invitrogen Neon Transfection System using Dharmacon SMARTpool siGENOME siRNA constructs for mouse *Mertk* (Catalog ID M-040357-00-0005) and *Tyro3* (Catalog ID M-043798-01-0005). Controls were generated by transfection with siGENOME Non-Targeting Control siRNA #2 (Catalog ID D-001206-14-05). Efficient knockout was confirmed by RT-qPCR.

For in vitro experiments, R992 (gift from RIGEL, South San Francisco, USA) was dissolved in DMSO and used in different concentrations as indicated. Recombinant plasma purified PROS1 (Enzyme Research Laboratories) was dissolved in water according to the manufacturer's instructions and used in different concentrations as indicated.

## Immunofluorescence

Cells were cultured on fibronectin (Sigma) coated glass coverslips for 48 h, fixed for 10 min in 4% PFA and permeabilized for 3–5 min with 0.01% TRITON X-100. Blocking was performed for 30 min with 1% BSA in PBS with Tween (PBS-T).

F-actin was visualized by incubating the cells with Alexa Fluor 488 Phalloidin (Invitrogen) (1:50) in a blocking solution for 30 min at room temperature in the dark.

Anti-Vinculin antibody V4139 (Sigma-Aldrich) was incubated as the first antibody (1:500) for 1 h at room temperature in a blocking solution. After rigorous washing, secondary antibody goat anti-rabbit AlexaFluor 555 (Thermofisher) (1:500) was incubated together with Alexa Fluor 488 Phalloidin (Invitrogen) (1:50) and DAPI (1:500) in blocking solution for 1 h at room temperature in the dark.

Anti-pMLC2 (Ser19) antibody #3671 (Cell Signaling) was incubated as the first antibody (1:100) for 1 h at room temperature in a blocking solution. After rigorous washing, secondary antibody goat anti-rabbit AlexaFluor 555 (Thermo Fisher) (1:200) was incubated together with Alexa Fluor 488 Phalloidin (Invitrogen) (1:50) and DAPI (1:500) in blocking solution for 1 h at room temperature in the dark.

After rigorous washing, cells were mounted in Vectashield mounting medium (Vector Laboratories).

Confocal microscopy was conducted using the Leica TCS SP8 X microscope (Software Leica LAS X). Immunofluorescence staining intensity was measured in at least 100 cells per group using ImageJ.

## Migration

Cell migration was assessed by wound healing assays using the Oris Cell Migration Assay platform consisting of a 96-well plate with stopper barriers that create a central cell-free detection zone[55]. Osteoblasts were seeded out and after the cells were confluent, the stoppers were removed. After 16 h, the cells were fixed with 4% PFA and stained by crystal violet dye. The migrated cells in the detection zone were analyzed by determining the relative area using ImageJ.

## Cell spreading

Calvarial cells were transfected with the indicated siRNA and cultured in an osteogenic medium with ascorbic acid and ß-glycerophosphate. After 5 days, cells were trypsinized and seeded out on fibronectin-coated glass coverslips for 10 and 20 min. Adherent cells were fixed with 4% PFA and stained for F-actin and DAPI. Five random pictures were acquired and the area of each cell was quantified. For spreading experiments with tumor CM, calvarial cells were cultured with tumor CM diluted 1:2 in osteogenic medium for 48 h ± R992. Then cells were trypsinized and seeded out on fibronectin-coated glass coverslips with tumor CM ± R992. At the indicated time point, cells were fixed with 4% PFA and stained for F-actin and DAPI. Five random pictures were selected and the area of each cell was quantified.

## RNA isolation/RT-qPCR

RNA was isolated using Fisher Scientific Invitrogen PureLink RNA Kit. cDNA synthesis was performed using a Thermo Scientific First strand cDNA synthesis kit. Quantitative RT-PCR was performed using Taqman GeneExpression Assays (ThermoFisher) and Eppendorf MasterCycler technology. Assay IDs Mm07295861_m1, Mm00444557_g1, Mm00437221_m1, Mm01343426_m1, Mm00490378_m1, Mm00475834_m1, Mm04933803_m1, Mm00501584_m1, and Mm03413826_mH are for *Mertk*, *Tyro3*, *Axl*, *Pros1*, *Gas6*, *Alpl*, SP7, *Runx2*, and *Bglap*, respectively. *Gapdh* was used as a housekeeping gene (assay ID Mm99999915_g1). Copy numbers were calculated using the standard curve method for absolute quantification.

## Immunoprecipitation/SDS-PAGE

For the harvest, cells were washed in ice-cold PBS and lysed in RIPA Buffer (Thermo Fisher), containing PhosSTOP phosphatase inhibitor cocktail (Roche) and protease inhibitor cocktail (Roche). Equivalent amounts of protein were incubated with respective primary antibodies for 2 h, followed by incubation with protein G-Sepharose (Thermo Fisher) for 3 h. The beads were washed three times in the RIPA buffer, resuspended in the appropriate volume of RIPA containing Laemmli gel loading buffer, and subjected to SDS-PAGE. The proteins were electrotransferred to nitrocellulose membranes and blocked in 5% bovine serum albumin for 1 h. Blocked membranes were probed with primary antibodies overnight at 4 °C in the same buffer, followed by secondary antibody conjugated to HRP in blocking solution for 1 h shaking at room temperature. Bands were visualized using enhanced chemiluminescence (ECL) detection.

## Detection of RHOA activation

GTP-bound RHOA was analyzed by performing a pull-down assay with Cell Biolabs, Inc. Rho Activation Assay kit. Calvarial cells were transfected and cultured as described. Three days after osteogenic induction, cells were incubated in serum-free α-MEM medium for 5 h and then stimulated with human-purified Protein S (Enzyme Research Laboratories). Cells were collected at the indicated time points and processed following the manufacturer's instructions.

## ELISA

Mouse P1NP (Novus biologicals) and Trap5b (immunodiagnostic systems) levels were determined in peripheral blood using ELISA kits according to the manufacturer's instructions. Human Igλ concentration was assessed in blood plasma from myeloma-bearing mice according to the manufacturer's instructions (Human Lambda ELISA KIT; Bethyl Laboratories Inc, Montgomery, TX, USA).

## Flow cytometry

The bone marrow of myeloma-bearing mice was flushed and MM cells were stained for CD138 (BioLegend). Events were captured using BD FACS Fortessa.

## Histology

The mice were fixed with 4% paraformaldehyde for 48 h and the tibia was decalcified with EDTA. Paraffin blocks were cut into 5-μm-thick sections. Two non-serial sections of each bone were assessed. TRAP staining was performed for 30 min followed by nuclear counterstaining with hematoxylin. The number of osteoclasts and osteoblasts at the bone surface was measured using the osteomeasure system (Osteometrics). Osteoclast precursor cells per bone marrow area or tumor area was quantified using Osteomeasure system (Osteometrics).

## Analysis of bone formation rate

About 2 mg/ml Calcein (Sigma) and 2 mg/ml Demeclocycline (Sigma) were injected i.p. 7 and 2 days before sacrificing the animals. Tibia were collected and fixed in 4% paraformaldehyde (PFA) for 48 h. For analysis, tibiae were embedded in methyl methacrylate. Samples were cut using 5-μm sagittal sections. Quantitative bone histomorphometric

measurements were performed according to standard protocols using an OsteoMeasure system (Osteometrics).

## μCT

μCT was used for 3D analyses of long bones. Long bones of mice were analyzed using high-resolution μCT with a fixed isotropic voxel size of 10 μm (70 peak kV at X μA 400 ms integration time; Viva80 microCT; Scanco Medical). All analyses were performed on digitally extracted bone tissue using 3D distance techniques (Scanco Medical), as reported previously[56]. Region of interest (ROI) was defined manually by drawing contours in slices.

## Statistics and reproducibility

Data were means ± SEMs. Statistical significance was determined by a two-tailed unpaired *t*-test, unless otherwise stated. Survival analysis was carried out using the Kaplan–Meier function (Mantel–Cox-test/ log-rank test). Significant outliers were calculated and excluded from the analysis. All statistical analyses were performed using GraphPad Prism 5 software. All cell culture experiments were performed at least three times with similar results. For data presentation, a representative experiment was chosen and included in the manuscript.

## Reporting summary

Further information on research design is available in the Nature Research Reporting Summary linked to this article.

## Data availability

All data supporting the findings described in this manuscript are available in the article and in the Supplementary Information and from the corresponding author upon reasonable request. Source data are provided with this paper.

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

## Acknowledgements

The position of J.E. was supported by the priority program μbone from the DFG (LO1863/5-1) and he received the E.W. Kuhlmann scholarship from the University Cancer Center Hamburg (UCCH). S.L. was supported by a Heisenberg professorship (DFG), by the priority program μbone from the DFG (LO1863/5-1), and is currently supported by the European Research Council (ERC) under the European Union's Horizon 2020 research and innovation program (Grant Agreement No. 758713), by the priority program μbone from the DFG (LO1863/5-1) and by the Hector Stiftung II. I.B.-B. was supported by (BE6658/1-1) and is currently supported by the priority program μbone from the DFG (BE6658/2-1). E.H. and H.T. received funding from the German Research Foundation (SPP 2084 to E.H. (HE 5208/5-1) and H.T (TA 1154/2-1) and Emmy Noether Program to H.T. (TA 1154/1-1 and TA 1154/1-2)). R992 was obtained from Rigel Inc. (South San Francisco, CA, USA). The authors would like to thank Michael Horn and the UCCH in vivo optical imaging core facility for help with the intracardial injections and imaging of our mouse models. Confocal microscopy was conducted at the UKE Microscopy Imaging Facility (DFG Research Infrastructure Portal: RI_00489). FACS analysis with BD Fortessa was performed at the FACS Sorting Core Unit at University Hospital Hamburg-Eppendorf. We thank Bao-Uyen Huynh and Sabrina Noster on behalf of the Animal Facility (FTH) at University Hospital Hamburg-Eppendorf for taking care of the mice housing and breeding.

## Author contributions

J.E., J.Z., V.G., N.B., T.V.L., M.E.V.D., A.B.-H., S.B., I.S.D., and E.H. performed experiments. K.R., E.M., T.B.-C, E.J.A., S.G., C.R., E.H., and H.T. provided animals, techniques, and intellectual feedback H.T., J.E., K.P., C.B., I.B.-B., and S.L were involved in data interpretation, in the writing, review, and/or revision of the manuscript. J.E., I.B.-B., and S.L. conceived the project, supervised the research, and wrote the original manuscript.

## Funding

## Competing interests

S.B., I.S.D., and E.M. are employees of Rigel Inc. S.L. has received speaker honoraria from Rigel Inc. The remaining authors declare no competing interests.

## Additional information

[1]Department of Oncology, Hematology and Bone Marrow Transplantation with Section Pneumology, Hubertus Wald Comprehensive Cancer Center Hamburg, University Medical Center Hamburg-Eppendorf, Hamburg, Germany. [2]Department of Tumor Biology, Center of Experimental Medicine, University Medical Center Hamburg-Eppendorf, Hamburg, Germany. [3]DKFZ-Hector Cancer Institute at the University Medical Center Mannheim, Mannheim, Germany. [4]Division of Personalized Medical Oncology (A420), German Cancer Research Center (DKFZ), Heidelberg, Germany. [5]Department of Personalized Oncology, University Hospital Mannheim, Medical Faculty Mannheim, University of Heidelberg, Mannheim, Germany. [6]Molecular Skeletal Biology Laboratory, Department of Trauma, Hand and Reconstructive Surgery, University Medical Center Hamburg-Eppendorf, Hamburg, Germany. [7]Institute of Musculoskeletal Medicine, University Hospital, LMU Munich, Martinsried, Germany. [8]Musculoskeletal University Center Munich, University Hospital, LMU Munich, Martinsried, Germany. [9]Department of Stem Cell Transplantation, Research Department Cell and Gene Therapy, University Medical Center Hamburg-Eppendorf, Hamburg, Germany. [10]Rigel Pharmaceuticals, Inc., South San Francisco, CA, USA. [11]Faculty of Dental Medicine, Institute for Dental Sciences, The Hebrew University of Jerusalem, Jerusalem, Israel. [12]Department of Immunobiology, Yale University School of Medicine, New Haven, CT, USA. [13]Department of Pharmacology, Yale University School of Medicine, New Haven, CT, USA. [14]Department of Neurology, Yale University School of Medicine, New Haven, CT, USA. [15]These authors contributed equally: Isabel Ben-Batalla, Sonja Loges. ✉e-mail: isabel.benbatalla@dkfz-heidelberg.de; s.loges@dkfz-heidelberg.de

