## [Peer Review File · Nature Communications]

Reviewers' Comments:

Reviewer #1:

Remarks to the Author:

Engelman and colleagues (NCOMMS-21-16249-T) report on a biochemical, cell biological, and in vivo study to investigate effects of Mertk and Tyro-3 (two members of the TAM family of type I RTKs) and their endogenous ligands PROS1 and Gas6 during osteoblast function/differentiation and osteoanabolic bone formation. Authors first explore expression patterns of Mertk, Tyro3, PROS1, and Gas6 in calvaria osteoblastic cultures treated in culture with osteogenic medium (ascorbic acid, beta-glycerophosphate, and dexamethasone) to show increased mRNA expression of aforementioned genes, and then generate conditional KO of Mertk and Tyro3 in osteoblasts by crossing floxed Mertk/Tyro3 mice with a Col1a1-cre+ that is specific for osteoblasts. Authors show using the murine models that Mertk and Tyro3 cKO regulate bone formation and differentiation in an antagonistic manner (Mertk KO increases and Tyro3 KO decreases) and then characterize biochemical and cell biological distinctions of Mertk and Tyro3 focusing on actin and RhoA/ROCK. Further studies introduce a Mertk tyrosine kinase inhibitor (R992) to assess whether pharmacological inhibition of Mertk can phenocopy effects of Mertk cKO to increase bone nodule formation and increase osteoblast differentiation. Finally, authors utilize bone metastasis models (U266 and MDA-MB-231) to assess whether R992 can reduce tumor bone metastasis and antagonize osteolytic bone loss in cancer progression.

Overall, this is a technically well-conceived study, particularly the generation of cKO of Tyro-3 and Mertk in osteoblasts and the results showing antagonistic functions in anabolic bone homeostasis. While the manuscript comes across a bit disjointed in flow (conditional KO's and bone biology, cell biology and Rho/ROCK connection, and cancer metastasis), the results are novel and the paper should be of interest to this research field. However, several queries came up in the review that should be addressed before publication.

While it appears that Gas6 and PROS1 are induced during osteogenic differentiation (Fig. 1 and other figures), is it known whether these factors are functionally active. This point is brought up since it does not appear the cell biological experiments have been performed uniformly. For example, in Fig. 2a, 2b, 2c, and 2d, shRNA knockdown experiments and morphology experiments were performed in the absence of PROS1 or Gas6, while in 2f and 2g, RhoA activation experiments were performed with exogenous added PROS1.

Likewise, in Fig. 2k, 2j, 2m, what was the chemoattractant (PROS, Gas6, serum)?

More information is needed to explain Supplemental 6, 7, and 8. Again, it appears that Supplemental 6 and 8 showed ligand independent effects of Mertk and PROS1 knockdown, while effects in Supplemental Fig. 7 show ligand (PROS1) dependent effects. Similar issues noted in SFig. 8 and SFig 9.

The results in Fig. 3 and Fig. are potentially interesting and important but require clarification. In this capacity, authors show relative specificity of R992 compound as a Mertk inhibitor over Tyro3 in cell culture (testing with agonistic mAb). However, how does this translate to 60 mg/kg BID? Collectively, it seems authors missed an opportunity to do more convincing experiments in the cKO Mertk and cKO Tyro3 models in the bone osteolysis experiments. Without this data, the present R992 data is not so convincing.

The authors are encouraged to present a cartoon model describing the summary of their experiments.

Reviewer #2:

Remarks to the Author:

In this manuscript, Engelmann et al. examine the role of Mertk and Tyro3 in regulating osteoblast activity, concluding in particular that Mertk suppresses bone formation and that small molecular inhibitors of Mertk increase bone mass. Lastly, data is offered that this pathway may be relevant in

cancer induced bone loss as shown by reduced bone loss after administration of a MERTK small molecule inhibitor. Overall, the topic is of interest, as gain of function/high bone mass phenotypes are not common and identification and mechanistic exploration of such phenotypes has potential therapeutic relevance for many bone disorders. Many aspects of the experimental approach are sound, however there are some critical concerns that must be addressed. Chief among these is that the experiment in Fig 4 is currently very difficult to interpret given that MERTK inhibition appears to be impacting osteoblasts, the tumor lines used and osteoclast-lineage cells, making it not possible to determine if this result is related to the function of MERTK studied elsewhere in the manuscript (comment #7 below). A few other important issues are noted below.

1. The phenotype shown in Fig 1 is modest to the degree that it is somewhat borderline in terms of making a clear case for the importance of this pathway. Was 2.3kb Col1a1-cre the only cre line examined? This cre line is notorious for providing very weak phenotypes in comparison with other osteoblast lineage cre lines, especially osterix-cre or Prx1-cre. Dmp1-cre also often provides more robust phenotypes than col1a1-cre, though typically less robust than osx or prx1-cre lines. If one of these other widely available cre lines can be used, it is likely that this would provide an overall stronger case for the importance of this pathway.

2. Is cortical bone mass impacted in the Fig 1 studies?

3. It seems that static histomorphometry parameters are provided in Fig 1. Was dynamic histomorphometry also performed?

4. All kinase and RTK inhibitors only show a relative but not absolute selectivity, and the less than one log difference in IC50 between MERTK and TYRO3 does not indicate strong selectivity. Based on this, claims of R992 selectivity appear to be overstated. As it is appreciated that there is no such thing as a perfectly selective kinase inhibitor, it would be sufficient to just tone down the selectivity claims in the text and acknowledge the potential issues with off-target effects in the discussion.

5. The data on conditioned medium transfer from various tumor cells is interesting, but genetic data is needed to have greater confidence that the results obtained represent a specific effect of MERTK activation. The ideal method would be to use lentiviral cre deletion in MERTK fl/fl calvarial osteoblasts and repeat this study, but MERTK knockdown methods would be acceptable.

6. What is the direct substrate of MERTK and TYRO3 leading to RhoA/Rock activation?

7. A major issue with the U266 and MDA-BM-231 study is that it is not possible to ascribe any of the functions of MERTK/R992 to osteoblasts when R992 is likely to directly act to inhibit or kill these tumor cell lines. Similarly the possible effect of R992 on osteoclasts adds another potential confounding factor complicating experimental interpretation. It is important that a study of tumor effects on bone mass be conducted in mice with a conditional deletion of MERTK in the osteoblasts lineage to confirm that the effects of R992 are not primarily due to its effects directly on tumor cells or osteoclast precursors. It is appreciated that the studies with human cells were appropriately conducted in NSG mice that preclude crossing in MERTK alleles, but various syngenic B6 tumor lines impacting bone are available. A system where the contribution of MERTK in osteoblasts to tumor effects on bone mass can be clearly ascertained is necessary.

8. The idea that Tyro3 and MERTK are antagonistic appears to be overstated as some opposite effects are shown in terms of overall bone mass and RhoA activation, but no data demonstrates direct functional antagonism between these two receptors aside from noting differences in their ultimate phenotypic effects. Either the data supporting direct functional antagonism between these two receptors or a change in how this concept is presented and discussed is needed. Related to this, is antagonism of MERTK and Tyro3 a known relationship between these genes in other contexts?

GENERAL REMARK: Authors would like to thank all reviewers for the effort they put in reviewing this paper and for their insightful comments. As requested, we have addressed all concerns and critiques of the referees in a point-by-point response.

RESPONSE TO REVIEWER #1

Engelman and colleagues (NCOMMS-21-16249-T) report on a biochemical, cell biological, and in vivo study to investigate effects of Mertk and Tyro-3 (two members of the TAM family of type I RTKs) and their endogenous ligands PROS1 and Gas6 during osteoblast function/differentiation and osteoanabolic bone formation. Authors first explore expression patterns of Mertk, Tyro3, PROS1, and Gas6 in calvaria osteoblastic cultures treated in culture with osteogenic medium (ascorbic acid, beta-glycerophosphate, and dexamethasone) to show increased mRNA expression of aforementioned genes, and then generate conditional KO of Mertk and Tyro3 in osteoblasts by crossing floxed Mertk/Tyro3 mice with a Col1a1-cre+ that is specific for osteoblasts. Authors show using the murine models that Mertk and Tyro3 cKO regulate bone formation and differentiation in an antagonistic manner (Mertk KO increases and Tyro3 KO decreases) and then characterize biochemical and cell biological distinctions of Mertk and Tyro3 focusing on actin and RhoA/ROCK. Further studies introduce a Mertk tyrosine kinase inhibitor (R992) to assess whether pharmacological inhibition of Mertk can phenocopy effects of Mertk cKO to increase bone nodule formation and increase osteoblast differentiation. Finally, authors utilize bone metastasis models (U266 and MDA-MB-231) to assess whether R992 can reduce tumor bone metastasis and antagonize osteolytic bone loss in cancer progression.

Overall, this is a technically well-conceived study, particularly the generation of cKO of Tyro-3 and Mertk in osteoblasts and the results showing antagonistic functions in anabolic bone homeostasis. While the manuscript comes across a bit disjointed in flow (conditional KO's and bone biology, cell biology and Rho/ROCK connection, and cancer metastasis), the results are novel and the paper should be of interest to this research field. However, several queries came up in the review that should be addressed before publication.

1) While it appears that Gas6 and PROS1 are induced during osteogenic differentiation (Fig. 1 and other figures), is it known whether these factors are functionally active. This point is brought up since it does not appear the cell biological experiments have been performed uniformly. For example, in Fig. 2a, 2b, 2c, and

2d, shRNA knockdown experiments and morphology experiments were performed in the absence of PROS1 or Gas6, while in 2f and 2g, RhoA activation experiments were performed with exogenous added PROS1.

Reply: Authors would like to thank the reviewer for this important comment. We repeated the experiments investigating effects of *Mertk* and *Tyro3* knockdown via siRNA on osteoblast cell morphology in presence of PROS1. We decided to focus on PROS1 because it can only bind to MERTK and TYRO3 whose role in bone homeostasis represent the focus of the manuscript. Future work is necessary to dissect potential overlaps and differences between the effects of GAS6 which additionally binds to the AXL receptor and PROS1 in this context.

Our new experiments revealed that PROS1 increases F-actin intensity, induces stress fiber formation and consistently pRLC intensity in osteoblasts in control conditions (NT). These effects were inhibited by silencing of *Mertk* (RFig. 1a-e; SFig. 9a-e of revised manuscript). In contrast, treatment of *Tyro3*-silenced osteoblasts with PROS1 increased F-actin intensity, induces high stress fiber formation and increased pRLC intensity, indicating that PROS1-MERTK axis promotes actin polymerization and stress fiber formation, whereas PROS1-TYRO3 axis inhibits these effects (RFig. 1a-e; SFig. 9a-e of revised manuscript). We added these results to the revised manuscript and replaced the experiments where PROS1 was not used (Fig. 2a-e of the original manuscript). We moved the novel siRNA data, now complemented with PROS1 and formerly shown in Fig. 2 to the supplemental figures because Reviewer #2 requested to generate stable knockouts of MERTK and TYRO3 in osteoblasts (Fig. 1, 2 and 3 of the revised manuscript). This request led us to the decision to repeat the *in vitro* experiments previously utilizing siRNA with these cells. For that aim, we treated calvarial cells from *Mertk*^{flox/flox} and *Tyro3*^{flox/flox} mice by extracellular vesicles containing recombinant CRE recombinase (TakaraBio).

Efficient MERTK knockout was evaluated by SDS-PAGE (RFig. 2a; Fig. 1j of revised manuscript). In concordance with siRNA data we found that PROS1 inhibits matrix osteoblast mineralization in control conditions while inducing it in absence of *Mertk* receptor (RFig. 2b; Fig. 1k of revised manuscript). Consistent findings were obtained when measuring alkaline phosphatase (*Alpl*) mRNA expression on day 7 and Osteocalcin (*Bglap*) on day 21. PROS1 decreased expression of these osteoblast differentiation markers, whereas MERTK KO osteoblasts showed increased expression (RFig. 2c and d; Fig. 1l and m of revised manuscript). Notably, PROS1 could not decrease osteoblast differentiation marker in absence of MERTK (RFig. 2c and d; Fig. 1l and m of revised manuscript). Altogether, these results show that PROS1-MERTK axis inhibits osteoblast differentiation and matrix mineralization.

Next, we investigated the effects of PROS1 on osteoblast differentiation in presence and absence of *Tyro3*. First, we show almost complete knockout of TYRO3 upon addition of CRE recombinase to *Tyro3*^{flox/flox} osteoblast precursors (RFig. 2e; Fig 3i of revised manuscript). Addition of PROS1 decreased matrix mineralization in TYRO3 KO osteoblast cultures (RFig. 2f; Fig 3j of revised manuscript). TYRO3 KO osteoblasts showed decreased expression of *Alpl* and *Bglap* which was further reduced in absence of TYRO3 (RFig. 2g and h; Fig 3g and h of revised manuscript)

Altogether our data show that PROS1-MERTK axis inhibits osteoblast differentiation and matrix mineralization whereas PROS1-TYRO3 promotes these processes.

We decided to exchange the *in vitro* data showing the biological effects of siRNA mediated *Mertk* and *Tyro3* knockdown on osteoblast differentiation and mineralization (SFig. 6 and 7 of the original version) with similar experiments using MERTK and TYRO3 stable KO osteoblasts. In these experiments we also

consistently included PROS1 treatments. As *in vitro* osteoblast differentiation including matrix mineralization represents a long-term process over a period of 4 weeks, we think that using stable KO strongly improves the quality of the data. In the experiments using MERTK KO and TYRO3 KO osteoblasts we could demonstrate consistently with our observations upon siRNA-mediated knock-down that PROS1 increases stress fiber formation, F-actin intensity, pRLC and vinculin staining intensity in control conditions (RFig. 3a-d and 4a-d; Fig. 2a-d and Fig. 3m-p of the revised manuscript). Besides, MERTK KO osteoblasts showed decreased stress fiber formation, F-actin and pRLC intensity with abrogation of the PROS1-mediated effects (RFig. 3a-d; Fig. 4a-d of the revised manuscript). In contrast, TYRO3 KO osteoblasts showed increased stress fiber formation, F-actin and pRLC intensity (RFig. 4a-d; Fig. 3m-p of the revised manuscript).

We carried out two additional experiments showing the effect of PROS1 on osteoblast mineralization in presence of the MERTK inhibitor R992 and in presence of the ROCK- inhibitor Y23763 (RFig. 5a and d; Fig. 1n and Fig. 2g, respectively in the revised manuscript). These experiments show that PROS1 cannot inhibit osteoblast matrix mineralization in presence of R992 and Y23763. Concomitantly, we measured the expression of differentiation marker *Alpl* and *Bglap* in presence of PROS1 and R992, which was not shown before in the original manuscript (RFig. 5b and c; Fig. 1o and p). For more clarity we show the MERTK- and TYRO3 data in separate figures in the revised manuscript (MERTK: Fig 1, 2; TYRO3: Fig. 3 of the revised manuscript).

We included all novel methodology and findings into the revised manuscript (p. 3, 4 and 12).

2) Likewise, in Fig. 2k, 2j, 2m, what was the chemoattractant (PROS, Gas6, serum)?

Reply: Authors appreciate this insightful comment of the reviewer and apologize for not having explained the experimental conditions of the migration assay properly. We would like to inform the reviewer that the applied Oris cell migration assay resembles a wound healing assay with a central cell-free detection zone into which the cells migrate¹. In the initial version of the manuscript, we did not add exogenous PROS1 or GAS6 and the assay was carried out with osteoblast differentiation medium. We replaced the wording “migration assay” by “wound healing assay” and added this information to the Materials & Methods part (p. 13 of the revised manuscript). In addition, we performed additional wound healing assays upon addition of PROS1 with *Mertk* and *Tyro3* siRNA-silenced osteoblasts based on the comment of the reviewer. Here, we observed increased osteoblast migration induced by PROS1 in control conditions, which was abrogated in *Mertk*-silenced osteoblasts (RFig. 6a and b; SFig. 10a and b of the revised manuscript). In contrast, PROS1 treatment in *Tyro3*-silenced osteoblasts increased osteoblast migration (RFig. 6a and b; SFig. 10a and b of the revised manuscript). These results show that PROS1-MERTK axis induces osteoblast migration, while PROS1-TYRO3 axis inhibits migration. Consistently, analysis of focal adhesion protein vinculin revealed that focal adhesion formation was increased by PROS1 in *NTsiRNA* controls (RFig. 6c and d; SFig. 10c and d of the revised manuscript). In *Mertk*-silenced osteoblasts PROS1 could not increase vinculin staining intensity (RFig. 6c and d; SFig. 10c and d of the revised manuscript), indicating that PROS1-MERTK axis promotes osteoblast focal adhesion formation. *Tyro3*-silenced osteoblasts showed increased vinculin staining intensity which could be further increased by PROS1, indicating that PROS1-TYRO3 axis inhibits focal adhesion formation.

As we described in our response to comment #1, we repeated also these experiments with stable KO of MERTK and TYRO3.

Here, we observed consistent with our previous data increased osteoblast migration induced by PROS1 in control conditions, which was abrogated in MERTK KO osteoblasts (RFig. 7a and b; SFig. 5a and b of the revised manuscript). Analysis of focal adhesion protein vinculin revealed that focal adhesion formation was increased by PROS1. MERTK KO osteoblasts exhibited decreased vinculin staining intensity and PROS1 could not induce focal adhesions in absence of MERTK (RFig. 7c and d; SFig. 5c and d of the revised manuscript). Furthermore, PROS1 increased and loss of MERTK decreased the percentage of cells with a leading and a trailing edge suggesting that MERTK controls polarization of migrating osteoblasts (RFig. 7e; SFig. 5e of the revised manuscript). These results indicate that MERTK induces osteoblast migration and focal adhesion formation.

In TYRO3 KO osteoblasts migration and focal adhesion formation was increased and PROS1 could further enhance these processes in TYRO3 KO osteoblasts (RFig. 8a-d; SFig. 7a-d of the revised manuscript). In TYRO3 KO osteoblasts, cells containing a leading and a trailing edge were increased and PROS1 could further increase this phenotype (RFig. 8e; SFig. 7e of the revised manuscript).

These results suggest that PROS1-TYRO3 axis promotes low F-actin content in osteoblasts with decelerated migration leading to increased osteoblast function. We replaced the siRNA data with the stable knock-down data in the revised version of the manuscript. The new results showing migration and focal adhesion formation in MERTK KO and TYRO3 KO osteoblasts were included into the revised manuscript (p. 4 and 5, SFig. 5 and SFig. 7).

3) More information is needed to explain Supplemental 6, 7, and 8. Again, it appears that Supplemental 6 and 8 showed ligand independent effects of Mertk and PROS1 knockdown, while effects in Supplemental Fig. 7 show ligand (PROS1) dependent effects. Similar issues noted in SFig. 8 and SFig 9.

Reply: Authors would like to thank the reviewer for this question. As mentioned above, we decided to exchange the *in vitro* data showing the biological effects of siRNA mediated *Mertk* and *Tyro3* knockdown on osteoblast differentiation and mineralization (Extended data Fig. 6 and 7 of the original version of the manuscript) with similar experiments using MERTK and TYRO3 stable KO osteoblasts (Fig. 1k,l and 3j,k of the revised manuscript).

In addition, to the described results in Comment 1, we repeated the experiments investigating osteoblast spreading and polarization in *Mertk* and *Tyro3* silenced osteoblasts with addition of PROS1 (Extended data Fig. 8 and 9 of the original version of the manuscript).

PROS1 increased the percentage of cells with a leading and a trailing edge in control osteoblasts (RFig. 9a), whereas in *Mertk*-silenced osteoblasts PROS1 treatment could not increase osteoblast polarization. In *Tyro3*-silenced osteoblasts cells containing a leading and a trailing edge were increased and PROS1 could further increase this phenotype, indicating that PROS1-MERTK axis induces and PROS1-TYRO3 axis inhibits osteoblast polarization (RFig. 9a). We obtained similar results on osteoblast polarization in experiments using stable MERTK and TYRO3 KO (RFig. 7 and 8). However, PROS1 treatment did not significantly affect osteoblast spreading in our experimental setting (RFig. 9b). This data was included into the revised manuscript (p. 5 and 6, SFig. 5 and SFig. 7, SFig. 11).

4) The results in Fig. 3 and Fig. are potentially interesting and important but require clarification. In this capacity, authors show relative specificity of R992 compound as a *Mertk* inhibitor over *Tyro3* in cell culture (testing with agonistic mAb). However, how does this translate to 60 mg/kg BID?

Reply: Pharmacokinetics of R992 in C57BL/6J mice were measured previously. It was observed that treatment of mice with 60mg/kg BID led to trough concentrations of 260 ng/ml (~50 nM) which increased up to 2020 ng/ml (~400 nM) 0.5h-1h after oral gavage. In our cell culture experiments, we observed dose dependently decreased MERTK phosphorylation with 80 and 400 nM R992, which corresponds to the aforementioned dose levels in mice reached with 60mg/kg BID².

The conclusion that the observed effects of R992 *in vivo* are predominantly mediated via MERTK are additionally strengthened by the new findings that osteoblast-specific deletion of *Mertk* phenocopies the effects of R992 in the tumor context leading to increased bone mass while deletion of *Tyro3* has the opposite effect (please see below, RFig. 10, Fig. 6 of the revised manuscript). Thus, if R992 would lead to a relevant inhibition of TYRO3 in addition to blocking MERTK both effects would outweigh each other which is not the case. We included the information of R992 pharmacokinetics into the revised manuscript (p. 8 and 9).

5) Collectively, it seems authors missed an opportunity to do more convincing experiments in the cKO *Mertk* and cKO *Tyro3* models in the bone osteolysis experiments. Without this data, the present R992 data is not so convincing.

Reply: Authors would like to thank the reviewer for this valid comment which was also raised by Reviewer #2. In order to address this issue, we chose the syngeneic EO771 breast cancer model because intracardiac injection of EO771 breast cancer cells into C57BL/6J mice leads to bone metastases within 2 weeks³.

We injected luciferase transduced EO771 cells into *Mertk*^{flox/flox} and *Col1a1-cre*⁺;*Mertk*^{flox/flox} C57BL/6J mice, monitored metastatic spread by Bioluminescence Imaging and performed μ CT imaging after 12 days. Tumor load in hind limbs was slightly decreased in *Col1a1-cre*⁺;*Mertk*^{flox/flox} mice three days after injection, but was not significantly changed after one week (RFig. 10a and b; Fig. 6a and b of the revised manuscript). Analysis of metaphyseal cancellous bone by μ CT showed higher bone volume in EO771 tumor bearing *Col1a1-cre*⁺;*Mertk*^{flox/flox} mice in comparison to *Mertk*^{flox/flox} mice after 12 days (RFig. 10c-h; Fig. 6c-h of the revised manuscript). Histochemistry of TRAP/Hematoxylin staining indicates increased osteoblast numbers in EO771 tumor bearing *Col1a1-cre*⁺;*Mertk*^{flox/flox} mice, whereas osteoclast numbers were not changed (RFig. 10i-k; Fig. 6i-k). These data phenocopy the findings obtained with the MERTK inhibitor R992 showing that blockade of MERTK receptor increases bone mass and osteoblasts in the context of breast cancer-induced osteolytic bone disease in an osteoblast-specific manner. Therefore, MERTK represents a novel target candidate to improve cancer-induced osteolytic bone disease by stimulation of osteoblasts. Further research is warranted to investigate whether MERTK inhibition is of value in other osteopenic bone disease such as osteoporosis.

On the other hand injection of EO771 cells into *Tyro3*^{flox/flox} and *Col1a1-cre*⁺;*Tyro3*^{flox/flox} C57BL/6J mice led to slightly increased hind limb tumor load at day 3, whereas tumor load in the later timepoint was not

changed (RFig. 10l and m; Fig. 6l and m of the revised manuscript). Analysis of metaphyseal cancellous bone by μ CT showed decreased bone volume and trabecular number in EO771 tumor bearing *Col1a1-cre⁺;Tyro3^{flox/flox}* mice in comparison to *Tyro3^{flox/flox}* mice after 12 days (RFig. 10n-s; Fig. 6n-s of the revised manuscript). Histomorphometry of TRAP/Hematoxylin staining revealed decreased osteoblast numbers in EO771 tumor bearing *Col1a1-cre⁺;Tyro3^{flox/flox}* mice, whereas osteoclast numbers were not changed (RFig. 10t-v; Fig. 6t-v of the revised manuscript).

These data confirm that presence of MERTK and not TYRO3 in osteoblasts promotes breast cancer-induced bone loss. Furthermore, they phenocopy data obtained by R992 thereby adding additional evidence that targeting MERTK increases bone mass via a direct effect on osteoblasts. We included all novel findings and methodology into the revised manuscript (p. 8, 9, 11 and 12).

6) The authors are encouraged to present a cartoon model describing the summary of their experiments.

Reply: Authors would like to thank the reviewer for this helpful suggestion. Following the advice of the reviewer we have included a graphical abstract summarizing the main findings of the paper (RFig. 11; Fig. 7 of the revised manuscript).

RESPONSE TO REVIEWER #2

In this manuscript, Engelmann et al. examine the role of Mertk and Tyro3 in regulating osteoblast activity, concluding in particular that Mertk suppresses bone formation and that small molecular inhibitors of Mertk increase bone mass. Lastly, data is offered that this pathway may be relevant in cancer induced bone loss as shown by reduced bone loss after administration of a Mertk small molecule inhibitor. Overall, the topic is of interest, as gain of function/high bone mass phenotypes are not common and identification and mechanistic exploration of such phenotypes has potential therapeutic relevance for many bone disorders. Many aspects of the experimental approach are sound, however there are some critical concerns that must be addressed. Chief among these is that the experiment in Fig 4 is currently very difficult to interpret given that Mertk inhibition appears to be impacting osteoblasts, the tumor lines used and osteoclast-lineage cells, making it not possible to determine if this result is related to the function of Mertk studied elsewhere in the manuscript (comment #7 below). A few other important issues are noted below.

1) The phenotype shown in Fig 1 is modest to the degree that it is somewhat borderline in terms of making a clear case for the importance of this pathway. Was 2.3kb Col1a1-cre the only cre line examined? This cre line is notorious for providing very weak phenotypes in comparison with other osteoblast lineage cre lines, especially osterix-cre or Prx1-cre. Dmp1-cre also often provides more robust phenotypes than col1a1-cre, though typically less robust than osx or prx1-cre lines. If one of these other widely available cre lines can be used, it is likely that this would provide an overall stronger case for the importance of this pathway.

Reply: We would like to thank the reviewer for this insightful question. The 2.3kb Col1a1-cre was the only cre line examined in this study. Authors agree with the comment of the reviewer that this cre line can

provide weaker phenotypes compared to other osteoblast-targeting cre lines. However, the phenotype is present and we now demonstrate in addition to the previous findings that Col1a1-mediated deletion of *Mertk* in osteoblasts also augments bone mass and osteoblast numbers in the syngeneic breast cancer model EO771 (please see our response to comment #5). These data indicate that blockade of MERTK receptor increases bone mass and osteoblasts also in the context of breast cancer-induced bone loss in an osteoblast-specific manner. The utilization of other osteoblast lineage cre lines acting at different stages of osteoblast development might provide stronger phenotypes and will be subject for further studies. We included this information into the revised manuscript (p. 8 and 9).

2) Is cortical bone mass impacted in the Fig 1 studies?

Reply: To answer this question we performed midshaft evaluation of femur in *Col1a1-cre⁺;Mertk^{fllox/fllox}* and *Col1a1-cre⁺;Tyro3^{fllox/fllox}* mice. *Col1a1-cre⁺;Mertk^{fllox/fllox}* mice did not show significant changes in femoral cortical bone thickness, whereas *Col1a1-cre⁺;Tyro3^{fllox/fllox}* mice exhibited decreased thickness of femoral midshaft (RFig. 12a and b). The additional evaluation of another bone parameter (Ct.Th.) could strengthen our finding that TYRO3 exerts a bone anabolic function. On the other hand, *Col1a1-cre⁺;Mertk^{fllox/fllox}* mice did not show significant changes in cortical thickness. We conclude that MERTK exerts a more significant function on trabecular bone, than on cortical bone. Also, other authors observed, that especially with the osteoblast specific *Col1a1-cre* some genes when impacting BV/TV do not always affect all other measured parameters^{4,5}.

These data were included into the revised manuscript (p. 3 and 4, SFig. 4d and 6d, respectively).

3) It seems that static histomorphometry parameters are provided in Fig 1. Was dynamic histomorphometry also performed?

Reply: Based on the suggestion of the reviewer we analyzed bone formation rate in *Col1a1-cre⁺;Mertk^{fllox/fllox}* and *Col1a1-cre⁺;Tyro3^{fllox/fllox}* mice. Therefore, we performed Calcein Demeclocyclin double labeling of newly formed bone and analyzed tibia of 8-week-old mice. *Col1a1-cre⁺;Mertk^{fllox/fllox}* mice showed increased bone formation rate, whereas in *Col1a1-cre⁺;Tyro3^{fllox/fllox}* mice bone formation was decreased (RFig. 13a-d). These data were added to the revised manuscript (p. 3, 4, and 14, Fig. 1d and e, Fig. 3d and e).

4) All kinase and RTK inhibitors only show a relative but not absolute selectivity, and the less than one log difference in IC50 between MERTK and TYRO3 does not indicate strong selectivity. Based on this, claims of R992 selectivity appear to be overstated. As it is appreciated that there is no such thing as a perfectly selective kinase inhibitor, it would be sufficient to just tone down the selectivity claims in the text and acknowledge the potential issues with off-target effects in the discussion.

Reply: Authors would like to thank the reviewer for this comment. We toned down our statements concerning selectivity of R992 towards MERTK over TYRO3 and discuss pharmacokinetics and potential off-target effects in the discussion of the revised manuscript (p. 9).

5) The data on conditioned medium transfer from various tumor cells is interesting, but genetic data is needed to have greater confidence that the results obtained represent a specific effect of MERTK activation. The ideal method would be to use lentiviral cre deletion in MERTK fl/fl calvarial osteoblasts and repeat this study, but MERTK knockdown methods would be acceptable.

Reply: We thank the reviewer for this comment which has led to new experiments substantially strengthening our findings. As recommended, we performed stable knockout of *Mertk* and *Tyro3* in calvarial osteoblasts using the CreloxP system (TakaraBio). We utilized this system also to repeat many of the osteoblast cell biology experiments where we previously utilized siRNA mediated knockdown in order to confirm our original data (please see below and response to reviewer 1 comments 1, 2 and 3).

To address the question of the reviewer, we treated lentiviral cre-mediated MERTK KO osteoblast cultures with conditioned medium (CM) of MDA-MB-231, U266 and H460 cancer cells. Matrix mineralization was increased in MERTK KO osteoblasts treated with CM of all three tumor cell lines (RFig. 14a; SFig. 21a of the revised manuscript). Concomitantly, osteoblast differentiation marker *Alpl*, *Runx2* and *Osx* were increased (RFig. 14b-d; SFig. 21b-d of the revised manuscript). These results suggest that activation of MERTK in osteoblasts may contribute to tumor-induced osteoblast inhibition in myeloma, breast- and lung cancer bone metastasis. Wound healing migration assays showed that MERTK KO osteoblast cultures treated with cancer cell CM exhibited decreased motility, indicating that tumor cells induce osteoblast motility via MERTK which counteracts osteoblast bone forming activity (RFig. 14e-h; SFig. 21e-h of the revised manuscript). Analysis of F-actin cytoskeleton by confocal microscopy revealed that cancer cells promote increased stress fiber formation and F-actin content in control osteoblasts but not MERTK KO osteoblasts (RFig. 14 i-k; SFig. 21i-k of the revised manuscript). Furthermore, we showed that cancer cells promote osteoblast contraction via increased MERTK mediated RLC phosphorylation (RFig. 14i and l; SFig. 21i and l of the revised manuscript). These data suggest that tumor cells induce a cytoskeletal reorganization in osteoblasts leading to retraction via MERTK, which inhibits osteoblast differentiation and bone forming capacity by inducing osteoblast motility. Therefore, MERTK could represent a target to reduce cancer induced osteoblast inhibition during osteolytic bone disease. We included those novel findings into the revised manuscript (p. 8 and 9).

Our cell biology experiments with stable knockout of *Mertk* and *Tyro3* showed that PROS1-MERTK axis inhibits osteoblast differentiation and matrix mineralization whereas PROS1-TYRO3 promotes these processes as shown in response to comment 3 of Reviewer 1. We furthermore validated our previous results describing a cytoskeletal regulation of osteoblasts by MERTK as well as TYRO3 KO. We could again demonstrate that PROS1 increases F-actin, pRLC and vinculin staining intensity. In contrast, MERTK KO osteoblasts showed decreased F-actin, pRLC and vinculin staining intensity and PROS1 could not increase these markers in absence of MERTK (RFig. 3 and RFig. 7; Fig. 2a-d and SFig. 5a-e, respectively of the revised manuscript). TYRO3 KO osteoblasts showed as expected high F-actin, pRLC intensity and increased focal adhesion formation (RFig. 4 and RFig. 8; Fig. 3m-p and SFig 7a-e, respectively, of the revised manuscript).

6) What is the direct substrate of MERTK and TYRO3 leading to RhoA/Rock activation?

Reply:

Authors would like to thank the reviewer for this question based on which we performed additional experiments. Evaluation of RHOA signaling revealed that PROS1 dose dependently induced activated GTPγS-RHOA in osteoblasts (RFig. 15a; Fig. 2e of the revised manuscript). It is known that MERTK can bind SH2 domain proteins, particularly the vav-proto oncogene (vav) family of guanine nucleotide exchange factors (GEFs) for Rho-family GTPases^{6,7}. Previous data show that tyrosine phosphorylation of MERTK and VAV leads to VAV dissociation from MERTK and downstream GDP to GTP exchange of RHOA⁶. VAV3 and VAV1 are expressed only in mature osteoblasts, whereas VAV2 is present throughout osteoblast differentiation⁸, suggesting that these RHO-GEFS could be involved in the activation of RHOA by MERTK because it binds VAV proteins constitutively. As we observed biological effects of MERTK in all stages of calvarial cell culture, we hypothesized that VAV2 is a substrate of MERTK leading to RHOA/ROCK activation. We could demonstrate that blocking of MERTK phosphorylation by R992 promotes high MERTK-VAV2 protein interaction leading to low VAV2 phosphorylation levels and inhibition of RHOA activation (RFig. 15b; Fig. 2f of the revised manuscript).

These data demonstrate that PROS1-MERTK axis activates the RHOA/ROCK axis via VAV2. We included these novel data into the revised manuscript (p. 4).

7) A major issue with the U266 and MDA-BM-231 study is that it is not possible to ascribe any of the functions of MERTK/R992 to osteoblasts when R992 is likely to directly act to inhibit or kill these tumor cell lines. Similarly the possible effect of R992 on osteoclasts adds another potential confounding factor complicating experimental interpretation. It is important that a study of tumor effects on bone mass be conducted in mice with a conditional deletion of MERTK in the osteoblasts lineage to confirm that the effects of R992 are not primarily due to its effects directly on tumor cells or osteoclast precursors. It is appreciated that the studies with human cells were appropriately conducted in NSG mice that preclude crossing in Mertk alleles, but various syngenic B6 tumor lines impacting bone are available. A system where the contribution of Mertk in osteoblasts to tumor effects on bone mass can be cleanly ascertained is necessary

Reply: Authors would like to thank the reviewer for this valid comment which was also raised by Reviewer #2. In order to address this issue, we chose the syngeneic EO771 breast cancer model because intracardiac injection of EO771 breast cancer cells into C57BL/6J mice leads to bone metastases within 2 weeks³.

We injected luciferase transduced EO771 cells into *Mertk*^{flox/flox} and *Col1a1-cre*⁺;*Mertk*^{flox/flox} C57BL/6J mice, monitored metastatic spread by Bioluminescence Imaging and performed μCT imaging after 12 days. Tumor load in hind limbs was slightly decreased in *Col1a1-cre*⁺;*Mertk*^{flox/flox} mice three days after injection, but was not significantly changed after one week (RFig. 10a and b; Fig. 6a and b of the revised manuscript). Analysis of metaphyseal cancellous bone by μCT showed higher bone volume in EO771 tumor bearing *Col1a1-cre*⁺;*Mertk*^{flox/flox} mice in comparison to *Mertk*^{flox/flox} mice after 12 days (RFig. 10c-h; Fig. 6c-h of the revised manuscript). Histomorphometry of TRAP/Hematoxylin staining indicates increased osteoblast

numbers in EO771 tumor bearing *Col1a1-cre⁺;Mertk^{flox/flox}* mice, whereas osteoclast numbers were not changed (RFig. 10i-k; Fig. 6i-k). These data phenocopy the findings obtained with the MERTK inhibitor R992 showing that blockade of MERTK receptor increases bone mass and osteoblasts in the context of breast cancer-induced osteolytic bone disease in an osteoblast-specific manner. Therefore, MERTK represents a novel target candidate to improve cancer-induced osteolytic bone disease by stimulation of osteoblasts. Further research is warranted to investigate whether MERTK inhibition is of value in other osteopenic bone disease such as osteoporosis.

On the other hand injection of EO771 cells into *Tyro3^{flox/flox}* and *Col1a1-cre⁺;Tyro3^{flox/flox}* C57BL/6J mice led to slightly increased hind limb tumor load at day 3, whereas tumor load in the later timepoint was not changed (RFig. 10l and m; Fig. 6l and m of the revised manuscript). Analysis of metaphyseal cancellous bone by μ CT showed decreased bone volume and trabecular number in EO771 tumor bearing *Col1a1-cre⁺;Tyro3^{flox/flox}* mice in comparison to *Tyro3^{flox/flox}* mice after 12 days (RFig. 10n-s; Fig. 6n-s of the revised manuscript). Histomorphometry of TRAP/Hematoxylin staining revealed decreased osteoblast numbers in EO771 tumor bearing *Col1a1-cre⁺;Tyro3^{flox/flox}* mice, whereas osteoclast numbers were not changed (RFig. 10t-v; Fig. 6t-v of the revised manuscript).

These data confirm that presence of MERTK and not TYRO3 in osteoblasts promotes breast cancer-induced bone loss. Furthermore, they phenocopy data obtained by R992 thereby adding additional evidence that targeting MERTK increases bone mass via a direct effect on osteoblasts. We included all novel findings into the revised manuscript (p. 8 and 9).

8) The idea that Tyro3 and Mertk are antagonistic appears to be overstated as some opposite effects are shown in terms of overall bone mass and RhoA activation, but no data demonstrates direct functional antagonism between these two receptors aside from noting differences in their ultimate phenotypic effects. Either the data supporting direct functional antagonism between these two receptors or a change in how this concept is presented and discussed is needed. Related to this, is antagonism of Mertk and Tyro3 a known relationship between these genes in other contexts?

Reply: Authors would like to thank the reviewer for this important comment which helps us to avoid overstatements related to our data. The reviewer is correct that our data do not demonstrate interactions between MERTK and TYRO3 at the functional level. Therefore, we removed all statements regarding functional antagonism between these two receptors from the result and discussion sections (page 7 of the original manuscript). The (potential) direct interaction between MERTK and TYRO3 needs to be investigated in future studies and is not known so far.

References

1 Lineham, E., Tizzard, G. J., Coles, S. J., Spencer, J. & Morley, S. J. Synergistic effects of inhibiting the MNK-eIF4E and PI3K/AKT/ mTOR pathways on cell migration in MDA-MB-231 cells. *Oncotarget* **9**, 14148-14159, doi:10.18632/oncotarget.24354 (2018).

2 Holland SJ, Owyang AM, Yi S, Young C, Braselmann S, Frances R, Bagos A, Tai E, Siu S, Park G, Lau D, Duan M, Bhamidipati S, Kolluri R, Darwish I, Ding P, Yu J, Duncton M, Singh R, Masuda E, Payan DG, Small molecule inhibitors of the anti-inflammatory TAM receptor MerTK #4869 AACR 2016

3 Hiraga, T. & Ninomiya, T. Establishment and characterization of a C57BL/6 mouse model of bone metastasis of breast cancer. *J Bone Miner Metab* **37**, 235-242, doi:10.1007/s00774-018-0927-y (2019).

4 Cao, H. *et al.* Focal adhesion protein Kindlin-2 regulates bone homeostasis in mice. *Bone Res* **8**, 2, doi:10.1038/s41413-019-0073-8 (2020).

5 Lee, S. Y. *et al.* Controlling hypoxia-inducible factor-2alpha is critical for maintaining bone homeostasis in mice. *Bone Res* **7**, 14, doi:10.1038/s41413-019-0054-y (2019).

6 Mahajan, N. P. & Earp, H. S. An SH2 domain-dependent, phosphotyrosine-independent interaction between Vav1 and the Mer receptor tyrosine kinase: a mechanism for localizing guanine nucleotide-exchange factor action. *J Biol Chem* **278**, 42596-42603, doi:10.1074/jbc.M305817200 (2003).

7 Shelby, S. J., Colwill, K., Dhe-Paganon, S., Pawson, T. & Thompson, D. A. MERTK interactions with SH2-domain proteins in the retinal pigment epithelium. *PLoS One* **8**, e53964, doi:10.1371/journal.pone.0053964 (2013).

8 Faccio, R. *et al.* Vav3 regulates osteoclast function and bone mass. *Nat Med* **11**, 284-290, doi:10.1038/nm1194 (2005).

Response letter Figure 1

a, Confocal imaging of F-actin staining of osteoblast cultures and osteoblast single cell analysis silenced for *Mertk* and *Tyro3* and treated with TAM receptor ligand PROS1 (100nM) on glass cover slips. **b**, F-actin immunofluorescence intensity of *Mertk* and *Tyro3* silenced osteoblasts treated with PROS1 (n=3, mean of 50 measurements in 3 fields). **c**, Quantification of stress fiber containing cells (n=3, mean of 100 cells in 3 fields). **d**, Representative confocal of pRLC staining in *Mertk* and *Tyro3* silenced osteoblasts treated with PROS1. **e**, pRLC immunofluorescence intensity (n=3, mean of 50 measurements in 3 fields) (*p<0.05, **p<0.01, ***p<0.001, ****p<0.0001 unpaired t-test). Data represent mean with SEM.

Response letter Figure 2

a, Analysis of MERTK protein in osteoblast cultures from *Mertk^{flox/flox}* mice treated with recombinant CRE recombinase. **b**, Alizarin Red staining of MERTK KO calvarial cell cultures treated with PROS1 (100nM). **c,d** RT-qPCR analysis of *Alpl* (**c**) and *Bglap* (**d**) mRNA expression in MERTK KO osteoblasts (n=3) (*p<0.05 unpaired t-test). **e**, Analysis of TYRO3 protein in osteoblast cultures from *Tyro3^{flox/flox}* mice treated with recombinant CRE recombinase. **f**, Alizarin Red staining of TYRO3 KO calvarial cell cultures treated with PROS1 (100nM). **g,h** RT-qPCR analysis of *Alpl* (**g**) and *Bglap* (**h**) mRNA expression in TYRO3 KO osteoblasts (n=3) (*p<0.05 unpaired t-test). Data represent mean with SEM.

Response letter Figure 3

a-d, Confocal imaging of pRLC and F-actin staining of MERTK KO osteoblasts on glass cover slips treated with TAM receptor ligand PROS1 (100nM) **(a)**. F-actin intensity (n=3, mean of 50 measurements in 3 fields) **(b)**, stress fiber containing cells (n=3, mean of 100 measurements in 3 fields) **(c)**, and pRLC intensity (n=3, mean of 50 measurements in 3 fields) **(d)** was quantified (*p<0.05 and **p<0.01 unpaired t-test). Data represent mean with SEM.

Response letter Figure 4

a-d, Confocal imaging of pRLC and F-actin staining of TYRO3 KO osteoblasts on glass cover slips treated with TAM receptor ligand PROS1 (100nM) **(a)**. F-actin intensity (n=3, mean of 50 measurements in 3 fields) **(b)**, stress fiber containing cells (n=3, mean of 100 measurements in 3 fields) **(c)** and pRLC intensity (n=3, mean of 50 measurements in 3 fields) **(d)** was quantified (*p<0.05 and **p<0.01 unpaired t-test). Data represent mean with SEM.

Response letter Figure 5

a

R992	-	-	+	+
PROS1	-	+	-	+

AlizarinRed

b

d

Y27632	-	-	+	+
PROS1	-	+	-	+

AlizarinRed

a, Alizarin Red staining of wild-type calvarial cell cultures treated with PROS1 (100nM) and MERTK-inhibitor R992 (200nM). **b,c** RT-qPCR analysis of *Alpl* (**b**) and *Bglap* (**c**) mRNA expression (n=3) **d**, Alizarin Red staining of wild-type calvarial cells treated with PROS1 (100nM) and ROCK-inhibitor Y27632 (10µM) (*p<0.05 unpaired t-test). Data represent mean with SEM.

Response letter Figure 6

a,b, Wound healing migration assay of osteoblasts lacking *Mertk* and *Tyro3* and treated with PROS1 (100nM). Representative pictures (**a**) and quantification of the relative migration (**b**) (n=3). **c**, Detection of focal adhesion formation by vinculin immunofluorescence staining in osteoblasts lacking *Mertk* and *Tyro3* and treated with PROS1 (100nM). **d**, Vinculin staining intensity was quantified (n=3, mean of 50 measurements in 3 fields) (*p<0.05, **p<0.01, ***p<0.001, ****p<0.0001 unpaired t-test). Data represent mean with SEM.

Response letter Figure 7

a, b, Wound healing migration assay of osteoblasts lacking *Mertk*. Representative pictures (**a**) and quantification of the relative migration (**b**) ($n=3$) **c**, Detection of focal adhesion formation by vinculin immunofluorescence staining in osteoblasts lacking *Mertk* and treated with PROS1. White arrow pointing to vinculin-rich focal adhesions. **d**, Vinculin staining intensity was quantified ($n=3$, mean of 50 measurements in 3 fields). **e**, Cells with a leading and a trailing edge were quantified ($n=3$, mean of 100 cells in 3 fields) (* $p<0.05$ unpaired t-test). Data represent mean with SEM.

Response letter Figure 8

a, b, Wound healing migration assay of osteoblasts lacking *Tyro3*. Representative pictures (**a**) and quantification of the relative migration (**b**) (n=3) **c**, Detection of focal adhesion formation by vinculin immunofluorescence staining in osteoblasts lacking *Tyro3* and treated with PROS1. White arrow pointing to vinculin-rich focal adhesions. **d**, Vinculin staining intensity was quantified (n=3, mean of 50 measurements in 3 fields). **e**, Cells with a leading and a trailing edge were quantified (n=3, mean of 100 cells in 3 fields) (*p<0.05 and **p<0.01 unpaired t-test). Data represent mean with SEM.

Response letter Figure 9

a, Cells with a leading and a trailing edge were quantified (n=100 measurements in 3 random fields) **b**, Quantification of cell area after 20 min (n=50-100 measurements in 3 random fields) (n=3, mean of 100 cells in 3 fields) (*p<0.05, **p<0.01 unpaired t-test). Data represent mean with SEM.

Response letter Figure 10

a-k, Luciferase⁺ EO771 breast cancer cells were injected intracardially in *Col1a1-cre⁺;Mertk^{flox/flox}* mice. **a,b**, Analysis of tumor load by Bioluminescence Imaging after 3 (**a**) and 7 (**b**) days. **c-h** μ CT 3D reconstructions (**c**) and analysis of BV (**d**), BV/TV (**e**), Tb.N (**f**), Tb.Th (**g**) and Tb.Sp (**h**) of trabecular bone of metaphyseal proximal region of the tibia (n=20/22). **i**, Representative pictures of TRAP/Hematoxylin staining of bone metastasis of EO771 injected mice. **j,k**, Histomorphometric analysis of osteoblast number (Ob.N/B.Pm) (**j**) and osteoclast number (Oc.N/B.Pm) (**k**) (n=12/13). **l-v**, Luciferase⁺ EO771 breast cancer cells were injected intracardially in *Col1a1-cre⁺;Tyro3^{flox/flox}* mice. **l,m**, Analysis of tumor load by Bioluminescence Imaging after 3 (**l**) and 7 (**m**) days. **n-s**, μ CT 3D reconstructions (**n**) and analysis of BV (**o**), BV/TV (**p**), Tb.N (**q**), Tb.Th (**r**) and Tb.Sp (**s**) of trabecular bone of metaphyseal proximal region of the tibia (n=20/22). **t**, Representative pictures of TRAP/Hematoxylin staining of bone metastasis of EO771 injected mice. **u,v**, Histomorphometric analysis of osteoblast number (Ob.N/B.Pm) (**u**) and osteoclast number (Oc.N/B.Pm) (**v**) (n=12/13) (*p<0.05 and **p<0.01 unpaired t-test). Data represent mean with SEM.

Response letter Figure 11

MERTK blockade increases bone formation by osteoblasts

Schematic outline of the mechanism showing how MERTK-PROS1 axis exerts osteopenia via VAV2-RHOA pathway. Inhibition of this axis led to normalization of bone homeostasis by increasing osteoblast function.

Response letter Figure 12

a,b, Midshaft evaluation of *Col1a1-cre⁺;Mertk^{flox/flox}* (*Mertk^{flox/flox}*, n=8; *Col1a1-cre⁺;Mertk^{flox/flox}*, n=11) (**a**) and *Col1a1-cre⁺;Tyro3^{flox/flox}* (*Tyro3^{flox/flox}*, n=6; *Col1a1-cre⁺;Tyro3^{flox/flox}*, n=9) (**b**) female mice after 8 weeks (*p<0.05 unpaired t-test). Data represent mean with SEM.

Response letter Figure 13

a, b, Representative pictures of Calcein Demeclocyclin labeling of *Mertk^{flox/flox}* and *Col1a1-cre⁺;Mertk^{flox/flox}* female mice (**a**) and bone formation rate (**b**) after 8 weeks **c, d**, Representative pictures of Calcein Demeclocyclin labeling of *Tyro3^{flox/flox}* and *Col1a1-cre⁺;Tyro3^{flox/flox}* female mice (**c**) and bone formation rate (**d**) after 8 weeks (* $p < 0.05$ unpaired t-test). Data represent mean with SEM.

Response letter Figure 14

a, Photographs of Alizarin Red staining of MERTK KO osteoblast cultures incubated with normal osteogenic control medium or tumor conditioned medium (CM) of U266, H460 and MDA-MB-231 cancer cells. **b,c,d**, Analysis of *Alpl*, *Runx2*, and *SP7* mRNA expression in MERTK KO osteoblasts in the presence of tumor CM from U266 (**b**), H460 (**c**) or MDA-MB-231 (**d**) data was normalized to osteoblasts incubated with the respective tumor CM alone. **e**, Photographs of wound healing migration assays of MERTK KO osteoblast cultures incubated with tumor CM of U266, H460 and MDA-MB-231 cancer cells. **f,g,h**, Analysis of migration in MERTK KO osteoblasts induced by tumor CM from U266 (**f**), H460 (**g**) or MDA-MB-231 (**h**), (n=3). **i**, Confocal imaging of F-actin and pRLC staining of MERTK KO osteoblasts on fibronectin coated glass cover slips. MERTK KO osteoblasts were incubated with tumor conditioned medium (CM) of U266, H460 and MDA-MB-231 cells for 48h. **j**, Stress fibers induced by tumor CM +/- R992 were quantified (n=100 measurements in 3 fields). **k**, F-actin immunofluorescence staining intensity was measured (n=50 measurements in 3 fields). **l**, pRLC immunofluorescence staining intensity was quantified (n=50 measurements in 3 fields) (*p<0.05, **p<0.01 unpaired t-test). Data represent mean with SEM.

Response letter Figure 15

a, Immunoblot of wild-type osteoblasts treated with different concentrations of PROS1 (0, 50, 100, 200 nM) showing activated GTP γ S-bound RHOA, total-RHOA and β -actin. **b**, Immunoblot of wild-type osteoblasts treated with PROS1 (100nM) and MERTK inhibitor R992 (200nM) showing activated GTP γ S-bound RHOA, total-RHOA, phosphorylated VAV2, association of MERTK and VAV2 and β -ACTIN. Data represent mean with SEM.

Reviewers' Comments:

Reviewer #1:

Remarks to the Author:

This is a revised manuscript that was previously review for novelty and technical merit. The authors have significantly revised the paper and addressed my queries, particularly whether their results underscored ligand inducible TAM activation. Authors have satisfactorily addressed my queries and concerns with solid new data. No further experiments are required from this reviewers perspective.

Reviewer #2:

Remarks to the Author:

Overall, the revised manuscript is responsive to the concerns raised in response to the initial manuscript. These issues have been adequately addressed and the resulting manuscript will be of interest for skeletal biology and the study of metastasis.